# Five dominant amino acid substitution signatures shape tumour immunity

Szilvia Juhász [ID] [1,2,8 ✉], Benjamin Tamás Papp [ID] [1,3,4,8], Anna Tácia Fülöp [ID] [1,3,5,6], Zoltán Farkas [ID] [1], Dávid Kókai[1,3], Dóra Alexandra Gyémánt[1,3,5], Franciska Tóth [ID] [1,3,6], Zsófia Nacsa [ID] [1], Dóra Spekhardt [ID] [1], Balázs Koncz[1,3], Péter Burkovics[7], Csaba Pál [ID] [1 ✉] & Máté Manczinger [ID] [1,3,4 ✉]

## Abstract

Although numerous mutational processes operate in cancer, their functional impacts are unclear. We hypothesised that certain mutation sources preferentially generate amino acid substitutions that evade immune recognition, producing immune-cold tumours regardless of tissue or mutation load. By analysing 9300 cancer exomes and performing mutagenesis experiments, we mapped links between mutagens, DNA-repair defects, and amino acid substitution signatures (AAS). Surprisingly, the spectrum collapsed into five recurrent AAS with distinct functional profiles. AAS4—generated by alkylating agents and mismatch-repair (MMR) deficiency and enriched in kidney and liver cancers—is less likely to accumulate hydrophobic residues, yielding poorly immunogenic neopeptides. These tumours display immune-desert microenvironments and respond poorly to immunotherapy. However, certain human leukocyte antigen (HLA) class I variants, such as HLA-B*07:02, correlate with immune-hot tumours in this subgroup. HLA-B*07:02, common in Europeans, presents proline-enriched neopeptides derived from AAS4 mutations. Supporting this, B*07:02-positive cancer cells harbouring AAS4-type mutations stimulated T-cell proliferation in vitro. These results show that neoantigen quality, not merely quantity, dictates antitumour immunity, explain inconsistent immunotherapy responses in MMR-deficient cancers, and advocate incorporating amino acid substitution patterns into predictive biomarkers and therapy design.

**Keywords** Cancer; Mutational Signatures; Mutations; Immunotherapy; Antitumor Immunity
**Subject Categories** Cancer; Computational Biology

## Introduction

Mutational signatures refer to the specific patterns of mutations (also known as mutational footprints) that arise in cancer genomes due to various mutagenic processes (Kucab et al, 2019; Helleday et al, 2014; Australian Pancreatic Cancer Genome Initiative et al, 2013; Nik-Zainal et al, 2015; Alexandrov et al, 2020). These processes might be driven by exposure to external agents like tobacco smoke (Alexandrov et al, 2016) or ultraviolet (UV) radiation (Hayward et al, 2017), internal factors like DNA repair deficiencies (Volkova et al, 2020; Zou et al, 2021) or the activities of specific enzymes (Alexandrov et al, 2020), and even certain biological processes such as aging (Alexandrov et al, 2020). Understanding these signatures can help to elucidate the history of mutagenic exposures and the processes driving a particular cancer's development. A deeper understanding of mutational signatures could also reveal therapeutic vulnerabilities of various tumours. For instance, identifying a signature linked to mismatch repair deficiency might suggest potential benefit from immune checkpoint inhibitors (Boiarsky et al, 2024; Gulhan et al, 2024).

The analysis of mutational signatures has become a basic component in cancer genomics. While this growth in the field is stunning, it is essential to consider potential constraints in its widening scope. Beyond a few pioneering studies focusing primarily on APOBEC3-related signatures (Boichard et al, 2019; Butler and Banday, 2023; DiMarco et al, 2022; Driscoll et al, 2020), the link between mutational signatures and cancer immunity has remained an unexplored area of research. Increased activity of APOBEC3 is a major source of mutations in multiple cancer types, like breast, bladder and non-small cell lung cancer (NSCLC) and has been associated with immune activation and response to immune checkpoint blockade (ICB) immunotherapy (Butler and Banday, 2023). However, it has remained unclear whether these mutagenic mechanisms linked with specific mutational signatures merely increase the number of neoantigens or produce uniquely potent neoantigens, especially primed to initiate anticancer immune responses. Recent research suggests the latter may be more accurate (Cummings et al, 2020). Melanomas, often tied to UV exposure, predominantly exhibit C-to-T base changes, which tend to produce peptides rich in negatively charged amino acids like glutamic acid. Notably, the HLA-B44 supertype preferentially binds peptides containing glutamic acid, potentially leading to enhanced survival rates in melanoma patients with B44 after ICB therapy (Cummings et al, 2020). Additionally, a clock-like mutational signature in

[1]Synthetic and Systems Biology Unit, Institute of Biochemistry, HUN-REN Biological Research Centre, Szeged, Hungary. [2]Cancer Microbiome Research Group, HCEMM, Szeged, Hungary. [3]HCEMM-BRC Systems Immunology Research Group, Szeged, Hungary. [4]Department of Dermatology and Allergology, University of Szeged, Szeged, Hungary. [5]National Academy of Scientist Education, Szeged, Hungary. [6]Doctoral School of Biology, University of Szeged, Szeged, Hungary. [7]Institute of Genetics, HUN-REN Biological Research Centre, Szeged, Hungary. [8]These authors contributed equally: Szilvia Juhász, Benjamin Tamás Papp. ✉E-mail: juhasz.szilvia@brc.hu; cpal@brc.hu; manczinger.mate@brc.hu

cancer samples from NSCLC and melanoma patients was associated with poor survival and lower immune activity (Chong et al, 2021), but the reasons remained unclear.

Inspired by prior studies (Cummings et al, 2020; Chong et al, 2021; Szpiech et al, 2017), here we comprehensively investigate the relationships between environmental mutagen exposure, HLA variations, and tumour immune characteristics. We hypothesise that both environmental mutagens and intrinsic cellular factors yield unique amino acid substitution patterns due to the mutation biases they induce. As a consequence, a wide range of environmental mutagens have characteristic and distinct impact on neopeptide formation and cancer immunity. Indeed, previous studies of microbial genomes have shown that nucleotide bias can significantly influence the amino acid makeup of the resulting proteins. As such, mutational bias can profoundly impact the molecular evolution of proteins (Bohlin et al, 2013; Li et al, 2015).

To investigate this hypothesis systematically, we present the concept of the amino acid substitution signature (AAS), which depicts specific patterns of amino acid modifications stemming from both external and internal factors. By analysing over 9000 cancer exomes, we identified five predominant AAS categories, encompassing various mutational signatures and cancer types. Our findings indicate that, based on the signature, AASs produce distinct types of neopeptides, thereby modulating cancer immune responses. We also pinpointed that certain AAS-HLA allele pairings are prime spots for immune recognition, irrespective of the tumour's origin or its mutation load. In essence, AASs serve as molecular indicators of tumour immunity and may be invaluable in forecasting patient outcomes.

## Results

### Characteristics of amino acid substitution signatures in tumour genomes

To identify dominant patterns of amino acid substitutions encoded by somatic mutations in cancer samples, we analysed 9374 tumour samples across 33 cancer types in The Cancer Genome Atlas (TCGA). By focusing on all non-synonymous mutations that have occurred in protein-coding genes, we computed the occurrence rate of every detected amino acid substitution ($n = 166$) in each sample, revealing significant diversity in frequencies (Appendix Fig. S1).

We employed the non-negative matrix factorisation (NMF) method to extract the most relevant signatures from the dataset (Fig. 1). NMF is an established tool for uncovering mutational patterns in cancer (Alexandrov et al, 2013). Our analysis led to the identification of five distinct amino acid substitution signatures (designated by AAS1 to AAS5, respectively, see Table EV1). Each signature is defined by characteristic, highly heterogenous distribution of amino acid substitution frequencies (Fig. 2B). In addition, we determined the prevalence of each AAS in cancer samples

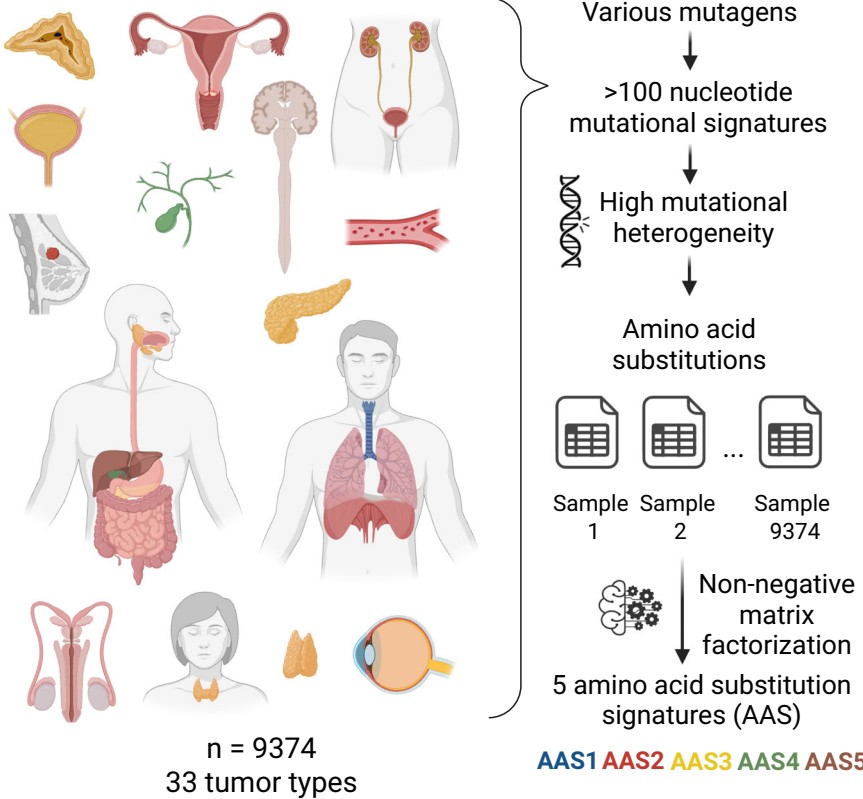

Figure 1. Amino acid substitutions in tumour samples can be categorised into five distinct signatures.

Overview of the dataset composition and methodological framework. See text of the 'Results' and 'Methods' sections for details.

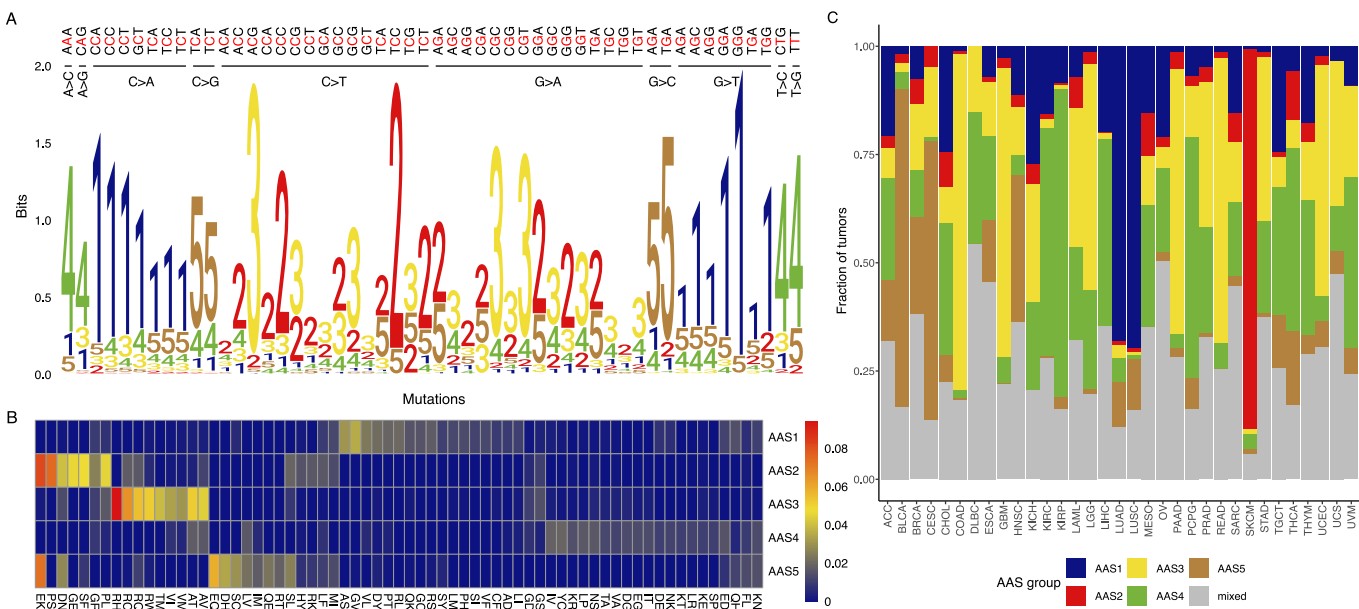

**Figure 2. Amino acid substitutions in tumour samples can be categorised into five distinct signatures.**

(A) Distinct AASs are linked to specific nucleotide substitution patterns. The motif illustrates how each nucleotide mutation in various 5′ and 3′ environments contributes to the formation of different AASs. For visualisation purposes, only nucleotide mutations reaching at least 0.5% in TCGA samples are shown (see Methods for details). (B) AASs are linked to specific substitution patterns. Values of the coefficient matrix acquired by NMF are used as a proxy of the association strength between different amino acid substitutions and each AAS. The heatmap indicates association strength values colour-coded. Only substitutions surpassing the value of 0.01 for at least one AAS are displayed for visualisation purposes. (C) A substantial portion of cancer samples is dominated by a single AAS. The plot illustrates the distribution of samples from various tumour types. Samples lacking a predominant AAS are categorised as 'mixed'. Source data are available online for this figure.

(Appendix Fig. S2). The cancer genomes are dominantly composed of a single signature, mixture of several signatures within cancer samples was found in 26% of samples (Fig. 2C). Using the COSMIC database (Tate et al, 2019) and prior experimental data on single-base substitution (SBS) signatures in human-induced pluripotent stem cells (Kucab et al, 2019; Zou et al, 2021), we additionally assessed associations between AASs, deficiencies in distinct DNA repair genes and a range of environmental mutagens (see "Methods", Fig. 3A; Appendix Figs. S3–S5). We summarised the characteristics of each AAS in Table EV1.

AAS1 is associated with C > A substitutions (G > T on the other strand, Fig. 2A), and is most commonly observed in lung cancer types, such as lung adenocarcinoma (LUAD) and squamous-cell carcinoma (LUSC) (Fig. 2C; Table EV1). There is a notable presence of AAS1 in cancer samples linked to tobacco carcinogens (including Benzo[a]pyrene, see Fig. 3A), marked by mutational signatures SBS4 and SBS29 (Fig. 3A). This suggests that AAS1 represents mutations stemming from the DNA damage caused by tobacco smoke mutagens. AAS1 is also linked to other environmental mutagens, including the chemotherapeutic agent platinum, cisplatin (SBS35), and aflatoxin (SBS24). The association of aflatoxin B1 and liver hepatocellular carcinoma (LIHC) with AAS1 is especially noteworthy, as dietary exposure to aflatoxin B1 is associated with an elevated incidence of hepatocellular carcinoma (Liu and Wu, 2010; Wu and Santella, 2012). In addition, AAS1-related mutations are associated with reactive oxygen species (ROS) induced DNA damage (SBS18) and potassium bromate (KBrO3). Potassium bromate is known to generate hydroxyl radicals, which induce a number of DNA lesions, with 8-oxo-dG

being the most abundant (Kawanishi and Murata, 2006). The corresponding mutations are typically repaired by base-excision repair pathways involving OGG1 and MUTYH (Zou et al, 2021).

AAS2 is characterised partly by C > T substitutions (Fig. 2A), and it is linked to glutamine to lysine (E > K), proline to serine (P > S), aspartate to asparagine (D > N), glycine to glutamate (G > E), substitutions (Fig. 2B). AAS2 traces back to multiple mutation mechanisms, including UV-light exposure (SBS7a and 7b) and deficiency in glycosylases involved in base-excision repair (NTHL1 and UNG). As might be expected, AAS2 is frequently observed in UV-induced melanoma (SKCM, Fig. 2C). AAS2 is also linked with SBS2, a substitution signature attributed to activity of the AID/APOBEC family of cytidine deaminases. SBS2-coupled APOBEC signatures exhibit high frequency of C > T transition at TpC, a pattern distinct from that exhibited by SBS13-coupled APOBEC signatures (C > G at TpN)(DiMarco et al, 2022).

AAS3 is strongly associated with arginine to histidine/cysteine/glutamine/tryptophan (R > H, R > C, R- > Q, R > W) substitutions (Fig. 2B). AAS3 is prevalent among tumours of the gastrointestinal tract, including colon (COAD), rectal (READ), pancreas (PAAD) cancer, stomach adenocarcinoma (STAD) (Fig. 2C). It can also be found in uterine corpus endometrial carcinoma (UCEC) and prostate cancer (PRAD) (Fig. 2C). Samples from these cancers frequently exhibit defects in the DNA mismatch repair mechanisms (SBS15 or SBS6). Defect of the mismatch repair gene PMS1 was particularly strongly linked to AAS3. Indeed, it produces substitution patterns related to SBS6, but distinct from those produced by other DNA mismatch repair genes (see AAS4, for details). In addition, AAS3 is associated with the SBS1 clock-like signature.

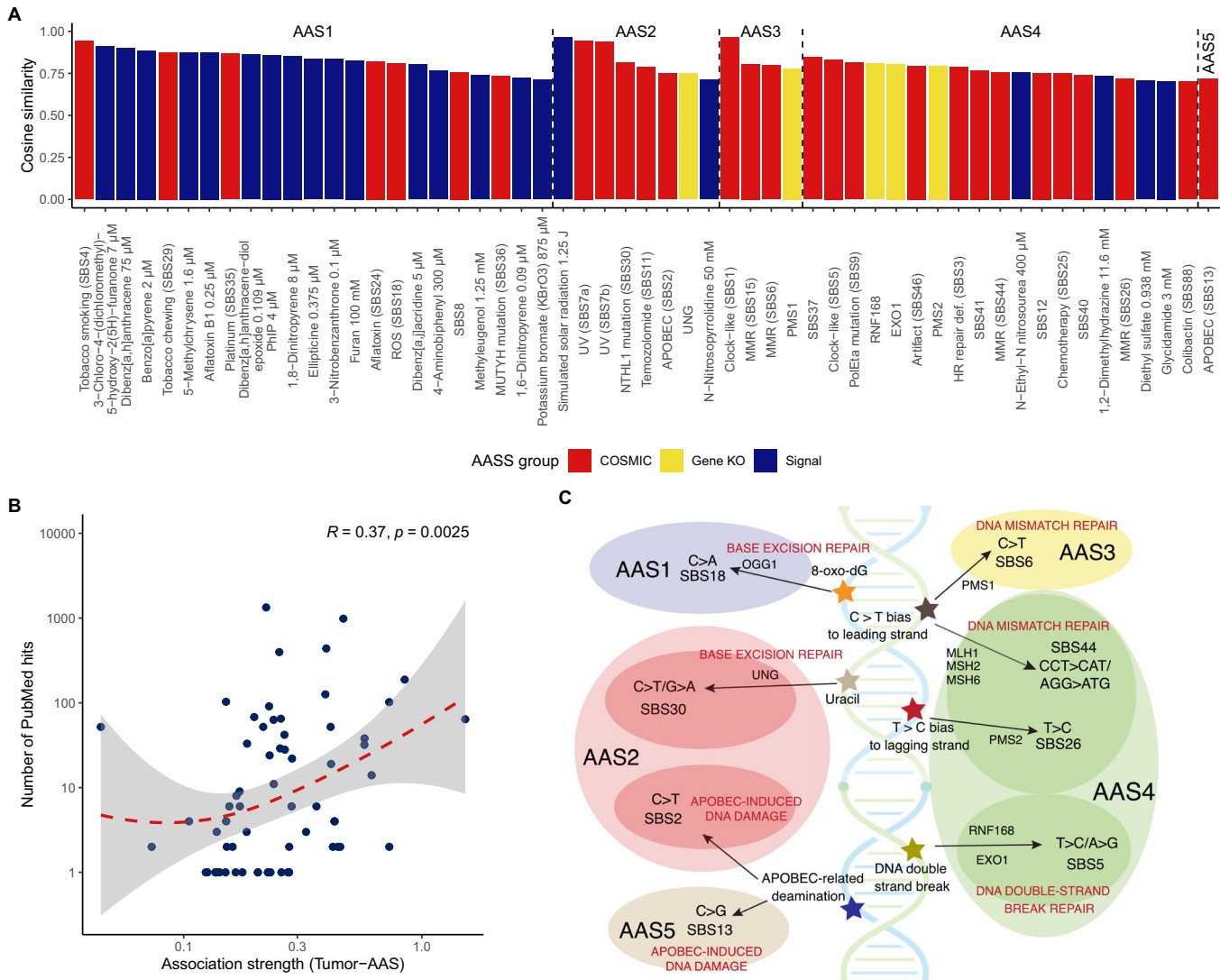

**Figure 3. Amino acid substitution signatures linked to environmental mutagens and endogenous mutational sources.**

(A) The strongest associations between mutagens and AASs. The cosine similarity between amino acid substitutions associated with different mutagenic processes and AASs. Only mutagens reaching higher than 0.7 cosine similarity to AASs are indicated. For each AAS, mutagens are ordered in decreasing order of cosine similarity. Red, yellow, and blue colours indicate the source of mutagen SBS data. (B) The association strength between AASs and tumour types is reflected by the number of PubMed hits. The horizontal axis indicates the median contribution of a given AAS to the mutation patterns in samples belonging to a given tumour type ($n = 66$ AAS-tumour type pairs). The vertical axis shows the total number of PubMed hits found for the tumour type and mutagens associated with the given AAS. Spearman's correlation coefficients and $P$ values of two-sided correlation tests are shown. The red dashed line indicates a smooth spline fitted with the locally estimated scatterplot smoothing (LOESS) method, while the grey-shaded areas indicate 95% confidence interval. Note that the vertical axis is log-transformed. (C) The figure illustrates the potential relationship between AASs and the deficiency of various DNA repair genes. Gene knockouts, affecting various DNA repair pathways, including mismatch repair (MMR), base-excision repair (BER), and double-strand break repair (DSBR), result in distinct mutational profiles (Zou et al, 2021). Cells with deletions in BER-related genes (UNG, OGG1) display specific AAS patterns: UNG-depletion is linked to AAS2, characterised by C > T transitions, while deletion of OGG1 is associated with AAS1, driven by C > A transversions. Similarly, DSBR-associated genes (EXO1, RNF168) both exhibit AAS4, caused by T > C substitutions. In contrast, MMR gene deficiencies show distinct patterns: lack of PMS1 aligns with AAS3, while lack of PMS2, MLH1, MSH2 and MSH6 signatures result in AAS4, mirroring known nucleotide mutational signatures (SBS6, SBS26, and SBS44) found in cancer samples (Alexandrov et al, 2020). These results highlight the key role of DNA repair genes in shaping mutational processes and amino acid substitution profiles, providing insights into cancer-associated mutagenesis. In addition, this figure illustrates the distinct mutational patterns of SBS2 and SBS13, both induced by APOBEC cytidine deaminases. SBS2 is characterised by C > T transitions, while SBS13 involves C > G transversions at TpN sites (Wang et al, 2023). These signatures frequently co-occur in cancer samples, highlighting the multifaceted role of APOBEC enzymes in cancer mutagenesis. Source data are available online for this figure.

AAS4 is associated with T > C/A > G and T > G/A > C substitutions (Fig. 2A), and it frequently appears in kidney cancers (KIRP, KIRC, Fig. 2C). AAS4 is linked to defects in several DNA mismatch repair genes, such as PMS2 (SBS26), MLH1, MSH2 and MSH6 (SBS44) (Fig. 3A; Appendix Fig. S5). In addition, AAS4 traces back to homologous recombination deficiency coupled mutational signatures, including loss of EXO1 and RNF168. Furthermore, alkylating agents, such as N-ethyl-N-nitrosourea (ENU) induce mutational signatures linked to AAS4. In line with this, several alkylating agents generate T > C transitions, which are the substrates of the aforementioned DNA repair genes (Kondo et al, 2010).

AAS5 is characterised mainly by C > G substitutions (Fig. 2A), frequently leading to glutamate to lysine/glutamine (E > K, E > Q, Fig. 2B) substitutions. AAS5 is mainly associated with the SBS13-coupled APOBEC signature. AAS5 reached high prevalence in bladder cancer (BLCA) and cervical cancer (CESC) samples. Reassuringly, the APOBEC system makes an important contribution to the mutational patterns in these cancer types (Butler and Banday, 2023).

In sum, we identified five main AAS classes that encompass diverse environmental mutagens and intrinsic mutational sources, characteristic of 33 different cancer types.

## Impact of environmental mutagens on the amino acid substitution landscape

The amino acid mutation profile in human cancers stems from a mix of uncontrollable environmental and internal exposures, all in highly variable genomic backgrounds. Consequently, a crucial next step involves examining the direct influence of environmental mutagens on amino acid substitution patterns. To this end, we studied previously described single-base mutational signatures in human-induced pluripotent stem cells that had been subjected to a wide array of environmental or therapeutic mutagens (Kucab et al, 2019). Using these data, we computationally evaluated the effects of 38 mutagens on the human proteome (see Methods). The frequency of all possible amino acid substitutions was evaluated and visualised on a heatmap across the mutagens studied (Appendix Fig. S6). The computational predictions were confirmed by additional mutational accumulation experiments in the laboratory. In particular, A549 cell lines were exposed to three environmental carcinogens, including Benzo[a]pyrene, N-ethyl-N-nitrosourea and UV-irradiation. Cell cultures were subjected to up to 70 cycles of mutagen exposure (Appendix Fig. S7), followed by whole-genome sequencing (WGS) and bioinformatic analyses of the mutated clonal cells ('Methods'). Between 75 and 3455 amino acid substitutions were detected in the sequenced samples, depending on the mutagen used (Appendix Table S1). Reassuringly, there was a strong positive correlation between the computationally predicted and experimentally observed frequencies of amino acid substitutions (Appendix Fig. S8).

Next, we explored the relationship between amino acid substitution frequencies in response to mutagens and the five delineated AASs (Fig. 3A; Appendix Fig. S3). Intriguingly, the five AASs are linked to distinct sets of environmental mutagens. In addition, functionally distinct mutagens produced similar amino acid substitution patterns, despite substantial differences in exact mutagenic effects. AAS1 is associated with aflatoxin, ochratoxin, furan, methyleugenol, PAHs and nitro-PAHs, heterocyclic amines, aromatic amines, ROS (KBrO3) and chemotherapeutic agents.

AAS2 is linked to UV and a nitrosamine, while AAS4 is associated with alkylating agents (e.g. nitrosourea) and glycidamide. Intriguingly, no exogenous mutagens were associated with AAS3 and AAS5. Our analysis indicates that intrinsic mutational sources, such as APOBEC-related mutagenesis (AAS5) and genetic deficiency in methyl-directed mismatch repair genes (AAS3) shape the formation of these two signatures (Fig. 3A; Appendix Figs. S4 and S5).

To systematically investigate the connections between AASs, environmental mutagens, and cancer types, we conducted a comprehensive text-mining analysis. This process involved querying PubMed for combinations of tumour types and mutagens, ensuring the search retrieved entries where both the tumour's full name and the mutagen or associated gene appeared in the record. We then quantified the number of PubMed hits linking specific tumour types with AASs, focusing on mutagens linked to the investigated AAS, as defined earlier. In addition, we analysed the contribution of each AAS to the genomic profiles of different tumour types, based on prior results (Appendix Fig. S2). Our findings demonstrated a positive correlation between the number of PubMed hits and the degree of association between tumour types and AASs (Fig. 3B). Practically, tumour type–AAS combinations showing stronger links were associated with a higher frequency of PubMed co-mentions connecting that tumour type to compounds or genes related to the AAS. This reveals that particular mutagens shape the amino acid substitution signatures in cancer genomes.

In sum, the above analyses show that five primary AAS classes encompass a broad spectrum of environmental mutagens. This suggests that despite significant differences in the underlying mutational processes, these diverse sources of mutations ultimately lead to similar patterns of amino acid substitutions, a finding with potential implications for cancer immunogenicity.

## Distinct amino acid substitution patterns associated with DNA repair pathways

To investigate the impact of different DNA repair pathways on AASs, we examined the mutational signature patterns observed in human pluripotent stem cells with deficiencies in various DNA replication and repair genes, as outlined in previous research. This study focused on key genes involved in DNA mismatch repair (MLH1, MSH2, MSH6, PMS1, PMS2), base-excision repair (UNG, OGG1) and double-strand break repair (EXO1, RNF168) (Zou et al, 2021). The authors generated isogenic knockouts of stem cells, cultured them without the addition of a mutagen, and performed WGS of selected subclones. The inactivation of each gene produced unique base and indel substitution signatures, thereby highlighting their essential role in promoting mutagenesis. We computationally assessed the mutagenic effects of each gene inactivation on the human proteome ('Methods'). This revealed distinct amino acid substitution patterns induced by each gene inactivation, which were then linked to the five AASs identified (Fig. 3C; Appendix Fig. S5).

DNA glycosylases perform a critical function in base-excision repair (BER), specifically in the identification and excision of damaged DNA bases (Krokan and Bjoras, 2013). Among these glycosylases, OGG1 and UNG are of particular significance. Polymorphisms in these genes have been investigated for potential associations with cancer susceptibility and progression with mixed results. OGG1 is responsible for the removal of the oxidative lesion

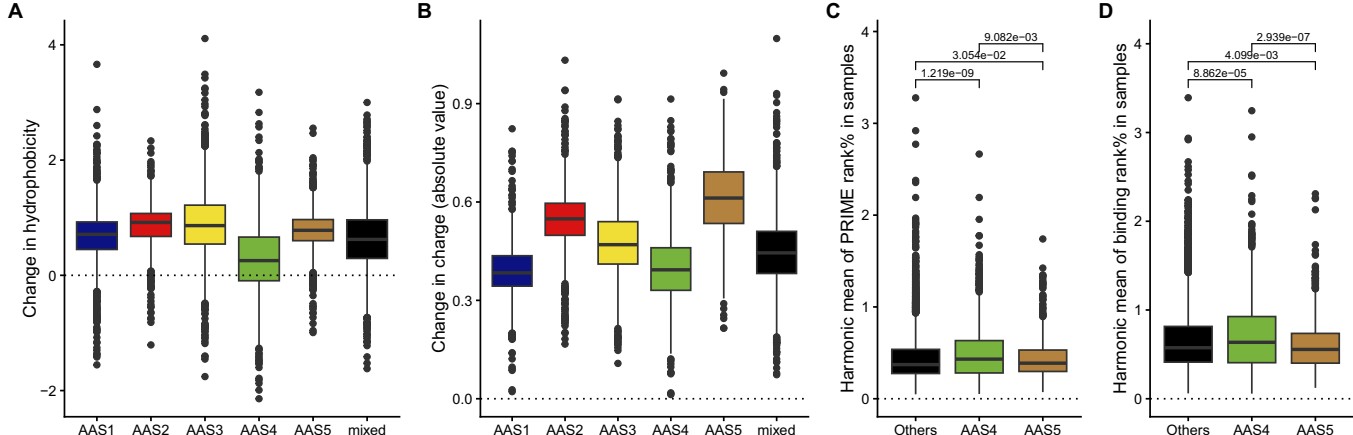

**Figure 4.  Amino acid substitution signatures influence neopeptide immunogenicity.**

(A, B) AASs are linked to diverse effects on the biophysical properties of amino acids. The effect of amino acid substitutions on hydrophobicity (**A**) and charge (**B**) in tumour samples dominated by different AASs ($n = 1388, 642, 2139, 1513, 1155,$ and $2537$ in AAS1 to 5 and mixed sample groups, respectively). Kruskal–Wallis rank-sum test $P$ values are lower than $2.2 \times 10^{-16}$ in both cases. FDR-corrected $P$ values of Dunn's test are not indicated due to visualisation purposes, they can be found in Appendix Table S2. (**C, D**) AAS4 and 5 are associated with the generation of low and high immunogenicity neopeptides, respectively. The harmonic mean of PRIME rank% (**C**) and HLA binding rank% (**D**) values are shown in tumour samples dominated by AAS4 ($n = 1379$ samples), AAS5 ($n = 1086$ samples) and other signatures ($n = 6116$ samples). Lower rank% values indicate higher immunogenicity (**C**) and stronger HLA binding (**D**). Kruskal–Wallis test $P$ values are $5.91 \times 10^{-5}$ and $8.7 \times 10^{-5}$ for (**C, D**), respectively. FDR-corrected $P$ values of Dunn's tests are indicated above horizontal segments. On boxplots, horizontal lines indicate median, boxes indicate the interquartile range (IQR) and whiskers indicate first quartile $-1.5 \times$ IQR and third quartile $+1.5 \times$ IQR. Source data are available online for this figure.

8-oxoguanine, which is caused by ROS (Ba and Boldogh, 2018). In contrast, UNG is involved in the excision of uracil from DNA (Zou et al, 2021). Despite both being involved in base-excision repair, the knockout profiles of ΔUNG and ΔOGG1 reveal substantial differences in their predicted amino acid substitution patterns (Fig. 3A; Appendix Fig. S5). ΔUNG is associated with AAS2, which is characterised by C > T substitutions, whereas ΔOGG1 is linked to AAS1, which involves C > A substitutions.

EXO1 and RNF168 are involved in DNA double-strand break repair (DSBR). EXO1 encodes a 5′ to 3′ exonuclease (Symington and Gautier, 2011), while RNF168 encodes an E3 ubiquitin ligase (Doil et al, 2009). Variants of EXO1 have been associated with susceptibility to colorectal cancer (Wu et al, 2001). Despite their different molecular functions, the base substitution signature of ΔRNF168 is highly similar to that of ΔEXO1, in particular the T > C peaks at ATA > ACA and TTA > TCA (Zou et al, 2021). Both knockouts are associated with AAS4 (Fig. 3A; Appendix Fig. S5).

The DNA mismatch repair (MMR) pathway exhibits variations that result in distinct patterns of amino acid substitutions. Specifically, the ΔPMS1 is associated with AAS3, while other knockouts in this pathway, such as ΔMSH2 and ΔMSH6, are linked to AAS4. This differentiation is consistent with the observation that the ΔPMS1 produces single-base substitution signatures that are significantly different from those of other MMR genes. PMS1 is primarily involved in the later stages of the repair process. At the same time, MSH2 and MSH6 form a heterodimer complex (MutSα) responsible for recognising and binding to mismatched base pairs during DNA replication, thereby playing a critical role in the early detection of single-base mismatches. Consequently, PMS1 knockout leads to C > T (SBS6), whereas MSH2/MSH6 knockouts result in CCT > CAT/AGG > ATG (SBS44) substitutions (Edelbrock et al, 2013; Warren et al, 2007; Zou et al, 2021).

## Amino acid substitution signatures shape the tumour microenvironment

To extract functional implications of the characteristic amino acid substitutions, we aimed to characterise the physicochemical attributes of each signature. For this purpose, the 20 standard amino acids were characterised based on fundamental physico-chemical properties, including hydrophobicity and charge. Using these characteristics, we computed the effect of each amino acid replacement on these attributes across the five AASs.

We found that tumour samples linked to AAS4-type substitutions are less likely to accumulate hydrophobic amino acids than samples linked to any other AASs, whereas those associated with AAS5-type substitutions tend to accumulate radical amino acid changes, with substantial changes in charge (Fig. 4A,B). For example, the median frequency of glutamate-to-lysine/glutamine substitutions (E > K, E > Q) is 3.8% in all cancer genomes, while they appear 4.4 times more frequently in AAS5-related cancer genomes (median 17%). These substitutions swap a negatively charged amino acid for a positively charged or non-charged amino acid, with potentially important functional consequences. These systematic changes in amino acid physicochemical properties across AASs could shape the immunogenic potential of neopeptides (Fritsch et al, 2014; Sim and Sun, 2022). First, immunogenic neopeptides are reported to be marked by elevated hydrophobicity of T-cell receptor contact residues (Chowell et al, 2015; Schmidt et al, 2021). Second, they could arise from radical amino acid changes that alter charge and enable HLA binding (Cummings et al, 2020). Indeed, displaying peptides with negatively charged amino acid anchors was associated with improved survival in ICB-treated cancer patients (Cummings et al, 2020). In the subsequent analyses, we focus specifically on AAS4 and AAS5. For more

detailed information on links between amino acid substitution signatures and immune phenotype, see Appendix Fig. S9.

As described above, amino acid substitutions linked to AAS4 tend to gather less hydrophobic amino acids and minimal changes in charge. These considerations indicate that tumour genomes dominated by AAS4 signatures should accumulate less immunogenic neopeptides. By contrast, cancer samples associated with AAS5 are expected to accumulate a relatively high number of immunogenic neopeptides. To test this hypothesis, we applied the PRIME algorithm to identify immunogenic epitopes (Schmidt et al, 2021). PRIME predicts epitope binding to HLA class I (HLA-I) molecules and their associated propensity for T-cell receptor recognition. Specifically, PRIME was used to assess the HLA-binding affinities and immunogenic potential of neopeptides in cancer samples linked to each AAS. Consistent with expectations, neopeptides from AAS4-dominant samples displayed significantly lower HLA-binding affinity and reduced immunogenicity scores compared to other sample groups (Fig. 4C,D). Conversely, AAS5-dominant samples produced neopeptides with markedly higher HLA-binding values (Fig. 4D), likely driven by the pronounced alterations in charge associated with amino acid substitutions unique to AAS5 (Fig. 4B).

To validate the impact of AASs on presented neopeptides, we analysed immunopeptidomics data from 17 cancer samples and cancer cell lines (Nicholas et al, 2023; Marty et al, 2017; Shapiro et al, 2025; Hirama et al, 2021; Newey et al, 2019; Gloger et al, 2016; Koumantou et al, 2019; Chen et al, 2018; Kalaora et al, 2016; Bassani-Sternberg et al, 2017). The analysis revealed that AAS type influences the likelihood of a substitution being identified in cancer samples (see Appendix

Supplementary Text and Appendix Figs. S16 and S17). Next, we asked how these substitution signatures translate into broader tumour immune phenotypes within the microenvironment. We hypothesised that cancer samples linked to AAS4 may exhibit a relatively low occurrence of immune-enriched tumour microenvironments, while those associated with AAS5 may display a higher frequency of such environments. To test these predictions, we employed a transcriptome-based tumour microenvironment classification of all cancer samples in TCGA (Bagaev et al, 2021). Prior work showed that patients possessing an immune-enriched tumour microenvironment benefited the most from cancer immunotherapy (Bagaev et al, 2021; Ji et al, 2012). As anticipated, tumour samples associated with AAS5 showed enrichment of immune-enriched tumour microenvironments (Fig. 5A, Fisher's exact test $P = 0.002$), while those linked to AAS4 exhibited depletion of such environments (Fig. 5A, Fisher's exact test $P = 5.4 \times 10^{-4}$). It is important to establish that the relationship between AAS classes and tumour microenvironments is independent of the number of somatic mutations and tumour type. For this purpose, we employed multiparametric logistic regression models, simultaneously considering AAS class, tumour mutational burden (TMB) and tumour types as predictive variables. Reassuringly, the effects of AAS4 and AAS5 on immune-enriched tumour microenvironment remained significant (Appendix Fig. S10).

To further characterise the intratumoral immune states associated with AAS4 and AAS5, we utilised features of tumour immune phenotypes outlined in a prior study (Thorsson et al, 2018). The analysis revealed that tumour samples characterised with AAS4 usually have lower, while those with AAS5 have higher overall lymphocyte

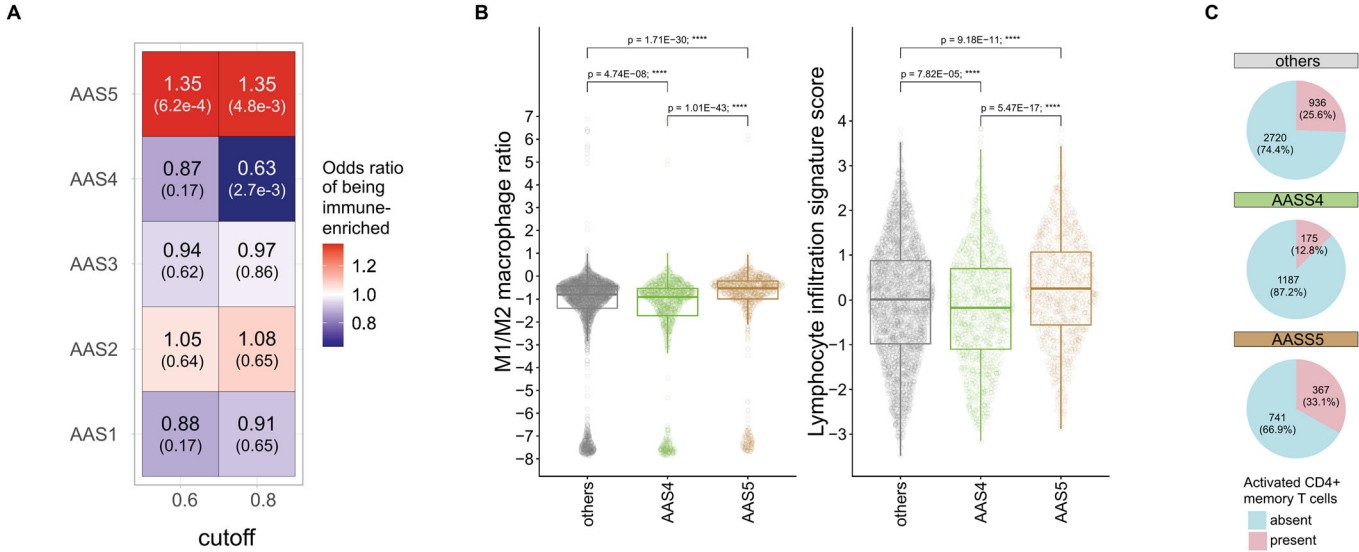

**Figure 5. Amino acid substitution signatures influence the tumour microenvironment.**

(A) The over- or underrepresentation of immune-enriched tumours in groups dominated by different AASs. Values are presented for two different cutoffs (0.6—$n = 1149$, 558, 1082, 1125 and 1053 for AAS1 to AAS5, respectively; 0.8—$n = 571$, 451, 472, 402 and 746 for AAS1 to AAS5, respectively) used for defining AAS-dominant groups. Each value represents the OR of finding immune-enriched tumour samples in groups dominated by the five different AASs vs. other groups. FDR-corrected p values of two-sided Fisher's exact tests are found in parentheses. (B) The logarithm of the ratio of M1/M2 macrophages (left), and lymphocyte infiltration signature score (right) of samples belonging to AAS4, AAS5 or other groups ($n = 1362$, 1108 and 3656 for M1/M2 macrophages, and 1361, 1107 and 3648 for lymphocyte infiltration score, respectively). We added $1 \times 10^{-8}$ to both M1 and M2 macrophage levels to prevent division by zero. Kruskal–Wallis test $P < 2.2 \times 10^{-16}$. On boxplots, horizontal lines indicate median, boxes indicate the IQR, and whiskers indicate first quartile $-1.5 \times IQR$ and third quartile $+1.5 \times IQR$. (C) The number and fraction of samples with activated memory CD4 + T cells among tumours dominated by AAS4, AAS5 or other AASs. OR (AAS4 vs. others): 0.43, Fisher's exact test P: $9.37 \times 10^{-24}$; OR (AAS5 vs. others): 1.44, Fisher's exact test $P$: $1.22 \times 10^{-6}$; OR (AAS5 vs. AAS4): 3.36, Fisher's exact test $P$: $1.01 \times 10^{-33}$. Source data are available online for this figure.

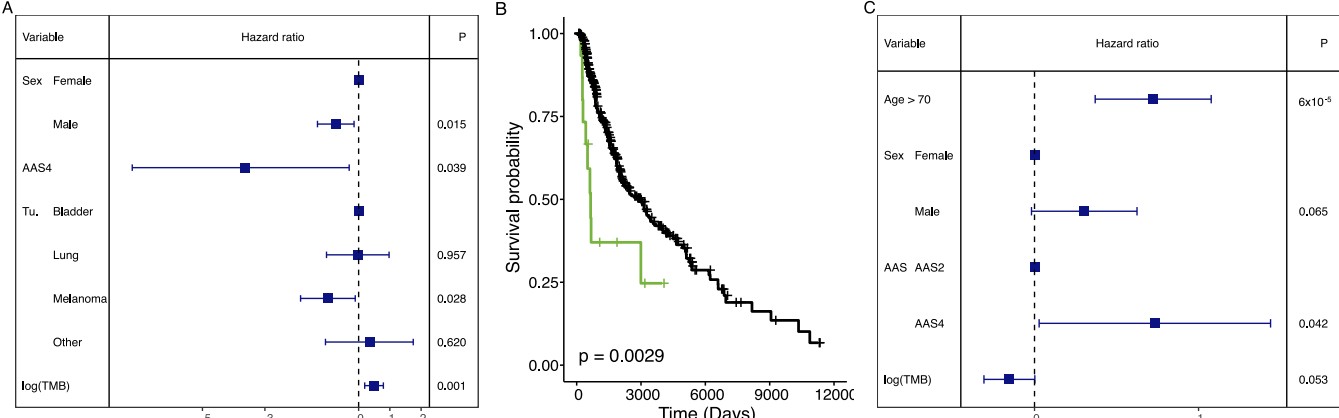

**Figure 6. Amino acid substitution signatures influence patient prognosis and response to cancer immunotherapy.**

(A) AAS4 is associated with low response rate to ICB therapy. The table shows the summary of a logistic regression model including sex ($n = 95$ females and 154 males), the logarithm of TMB, tumour type (Tu., $n = 27$, 57, 151 and 14 in bladder, lung, melanoma, and other tumour groups, respectively) and the dominance by AAS4 as a covariate. Blue squares indicate hazard ratios associated with each independent variable, while blue horizontal lines indicate 95% confidence intervals; $P$ values of two-sided Z statistics are shown. (B) AAS4-dominant melanoma samples are associated with worse survival. The Kaplan–Meier curve shows the overall survival probability of melanoma samples in the TCGA database dominated by AAS4 (green, $n = 16$ samples) or AAS2 (black, $n = 407$ samples). The $P$ value of a log-rank test is indicated. (C) The effect of AAS4 dominance on the overall survival of melanoma patients remains significant after controlling for age ($n = 96$ and $n = 296$ for patients aged >70 years and ≤70 years, respectively), sex ($n = 147$ females and 253 males) and the logarithm of TMB. The table shows the summary of a multiple Cox regression model. Blue squares indicate hazard ratios associated with each independent variable, while blue horizontal lines indicate 95% confidence intervals; $P$ values of two-sided Z statistics are shown. Source data are available online for this figure.

infiltration levels (Fig. 5B). This feature is generally linked to the efficacy of anticancer immune response. Recently, the ratio of M1 to M2 tumour-associated macrophages (TAM) within the tumour microenvironment emerged as a biomarker, where a relatively higher M1/M2 ratio indicates better prognosis in cancer patients (Jayasingam et al, 2020). As expected, AAS5-dominant tumours had higher M1/M2 ratio than AAS4-dominant ones (Fig. 5B). In addition, active CD4+ memory T cells play a crucial role in orchestrating and sustaining immune responses against cancer: Higher counts of active CD4+ memory T cells can enhance the capacity to recognise and attack cancer cells, leading to a more effective antitumor response (Beckhove et al, 2004; Ning et al, 2021). In line with the above results, active CD4+ memory cell number was particularly low in AAS4-associated tumours (Fig. 5C). These results remained significant in regression models involving TMB and tumour types as covariates. (Appendix Table S3).

## AAS4-dominated tumours are associated with a worse prognosis

Based on the above results, we next investigated the impact of amino acid substitution patterns on cancer immunotherapy. We focused on a previously published cohort of 249 cancer patients treated with immune checkpoint therapy across multiple cancer types, including melanoma, NSCLC and bladder cancer, among others (Miao et al, 2018). Most of these patients were treated with anti-PD-1, anti-PD-L1, anti-CTLA-4 therapies, or a combination thereof. Specifically, we analysed the mutational data of the corresponding tumour samples and clinically annotated outcomes to immune checkpoint therapy. We evaluated how amino acid substitution signatures affect clinical outcomes by applying the Response Evaluation Criteria in Solid Tumours (RECIST).

Consistent with earlier research (Roh et al, 2017), a clinical benefit from treatment was considered as either a complete or partial response or stable disease for ≥6 months. Conversely, no clinical benefit was categorised as stable disease for <6 months or progressive disease. We found that the level of AAS4 in cancer samples is negatively associated with clinical benefit from immune checkpoint treatments in a multivariate regression model, which contained tumour type, mutational burden and gender as covariates (Fig. 6A).

Next, we examined whether the prevalence of AAS4 in tumour genomes shapes disease progression in those patients who did not receive immune checkpoint therapy. The mutational landscape of melanoma is linked with diverse carcinogenic processes across its subtypes, some of which are unrelated to sun exposure (Hayward et al, 2017). Patients with non-UV-associated melanomas display worse prognosis than patients with UV-associated melanomas(Hayward et al, 2017), but the reasons are largely unknown. Here, we hypothesised that this pattern could be partly attributed to the prevalence of AAS4 in non-UV-associated melanoma genomes. The majority of the melanoma genomes in TCGA are dominated by AAS2 ($n = 407$), most likely reflecting the mutagenic effects of UV, while non-UV-associated melanoma genomic samples dominated by AAS4 are relatively rare ($n = 16$). In line with expectation, AAS4-dominant samples were associated with a significantly worse survival than AAS2-dominant ones (Fig. 6B). The effect remained significant in a multiple Cox regression model after controlling for age, gender, and TMB (Fig. 6C). Repeating the analysis with COSMIC SBS exposures in place of AASs did not improve model performance (see 'Methods', BIC: 1870.2 for the AAS model vs. 1879.9 for the SBS model; two-sided ANOVA $P = 0.11$), supporting the interpretation that AASs provide a robust, low-dimensional representation of mutational processes at the amino acid level.

While these patterns are intriguing, additional data will be needed to establish the prognostic effects of AAS4 on antitumor immunity.

Finally, it is important to interpret these findings considering intratumor heterogeneity (ITH) of mutations, which refers to the presence of multiple, distinct populations of cancer cells within a single tumour (Swanton, 2012). Specifically, we used a previously published dataset where ITH was defined as the fraction of mutations that are subclonal and present in a minority of tumour cells (Thorsson et al, 2018). Substantial intratumor heterogeneity in human cancers is associated with decreased T-cell infiltration and poor survival rates (Vitale et al, 2021). As a consequence, tumour heterogeneity poses significant challenges to effective anticancer immune responses and therapeutic strategies. Reassuringly, tumour samples associated with AAS4 typically exhibit lower levels of intratumor heterogeneity compared to tumours linked with other amino acid substitution signatures (Appendix Fig. S11). Hence, ITH is unlikely to account for the low immunogenicity observed in AAS4-type tumours, as this hypothesis would predict a positive correlation between AAS4 and ITH.

## Amino acid substitution signature and HLA-genotype jointly shape anticancer immune response

Neopeptides originate from intracellular proteins that undergo cleavage by the proteasome and peptidases (Manczinger et al, 2021). These peptides are then loaded onto the highly variable surface HLA-I proteins for presentation to cytotoxic T cells. Hence, the peptide-binding specificity of HLA variants influences the immune response by affecting which peptides can be presented to cytotoxic T cells. Given the exceptionally high diversity in HLA binding preferences across human populations (Robinson et al, 2024), we hypothesised that specific HLA-I variants exist that can efficiently recognise neopeptides characterised by unusual amino acid substitution patterns. We focused on tumour samples associated with AAS4, as these tumours were generally, but not always, characterised by a lymphocyte-depleted tumour microenvironment and an ineffective antitumor immune response (Fig. 5).

Our primary objective was to assess the specificity of individual HLA-I alleles for the 20 amino acids using a large-scale immunopeptidomics dataset (Sarkizova et al, 2020) (see 'Methods'). We analysed over 185,000 experimentally verified peptide–HLA interactions, encompassing 92 HLA-A, -B, and -C variants supported by appropriate in vitro data. These alleles are detectable at significant frequencies and represent a substantial portion of individuals across various human populations. Notably, our analysis included all HLA-A and B alleles from a widely recognised reference set that offers maximal population coverage (Weiskopf et al, 2013). We determined the amino acid composition of peptides, ranging from 8 to 12 amino acids in length, bound by the peptide-binding domains of each HLA-I variant ('Methods'). Based on these data, we calculated the binding specificity of each HLA variant to each amino acid.

Several mutated amino acids associated with AAS4 substitutions, such as cysteine (C), valine (V), threonine (T), and serine (S), are generally poorly presented by most HLA-I variants (Fig. 7B; Appendix Fig. S12). However, we found that immune-enriched tumours (estimated by the lymphocyte infiltration level) were especially likely to carry HLA variants with enhanced capacity to present AAS4-specific mutant peptides (Fig. 7A). In particular, HLA-B*27:05, HLA-B*07:02 and HLA-B*40:01 are enriched in AAS4-associated tumour

samples with an immune-enriched microenvironment. These HLA variants possess distinctive structural characteristics within their peptide-binding grooves, which allow them to efficiently present peptides containing specific amino acids linked to AAS4 (Fig. 7B; Appendix Fig. S12). For example, HLA-B*27:05, HLA-B*07:02, and HLA-B*40:01 exhibit an enhanced ability to present peptides that include arginine (R), proline (P), and glutamate (E), respectively.

Building on these observations, we next investigated whether the interaction between AAS class and HLA genotype also modulates immune selection across tumours. We classified tumours as immune-selected as previously, using an analysis of substitution patterns as a quantitative measure of immune-mediated selection in cancer genomes (Zapata et al, 2023). To assess the joint influence of AAS class and HLA binding properties, tumours were further stratified according to the predicted affinity of their HLA alleles for the amino acid substitutions characteristic of their assigned AAS class. Multivariate logistic regression models were then constructed with immune selection status as the dependent variable and AAS class, tumour mutational burden, and HLA affinity group as independent predictors. The analysis revealed that tumours belonging to AAS1 and AAS5 displayed the strongest signatures of immune selection (Appendix Fig. S13), whereas AAS2–AAS4 exhibited comparatively weaker associations. Consistent with expectations, higher HLA affinity correlated positively with immune selection. Collectively, these findings highlight that both AAS class and HLA genotype jointly determine the intensity and nature of immune selection in cancer evolution.

## HLA-B*07:02-carrying cancer cells induce T-cell proliferation after AAS4-associated mutagenesis

In this proof-of-concept analysis, we focused on ENU, a potent alkylating agent commonly used in cancer research for its ability to induce mutagenesis by transferring ethyl groups to DNA bases (Noveroske et al, 2000). This process creates genetic lesions that disrupt DNA replication and cell division. ENU is generally associated with AAS4 (Fig. 3A; Appendix Fig. S3). Most common HLA-I variants are unlikely to present peptides generated by ENU-induced mutations as the most common mutated amino acids are alanine (A), glycine (G) and proline (P) (Appendix Fig. S8). Neopeptides containing these amino acids are generally poorly bound by HLA-I (Appendix Fig. S12). However, proline-rich peptides are especially likely to be presented by certain HLA-B molecules, surpassing the peptide presentation efficiency of most other HLA-I variants (Appendix Fig. S12). In specific, HLA-B*07:02, present in up to 20% of human populations (Gonzalez-Galarza et al, 2019), is specific for proline (P) at the second position of peptides, and proline frequently replaces serine (S) and leucine (L) following ENU treatment (Appendix Fig. S8). Consequently, this mutagen is expected to generate mutant cells that could elicit an anticancer immune response in individuals carrying HLA-B*07:02.

To investigate the combined effect of mutagens and HLA variation on immune cell proliferation (Fig. 7C), we first performed an in-depth mutation analysis (Appendix Table S1) using a genetically modified cell line carrying only HLA-B*07:02 molecules within an A549 lung carcinoma cell background. For control, we used isogenic cell lines expressing only HLA-A*03:01, which is specific for lysine (K, rarely mutated after ENU treatment, see Appendix Figs. S9 and S12), or lacking HLA-I molecules entirely (HLA KO) (see "Methods"). After 70 cycles of ENU exposure, all samples showed a high mutation count,

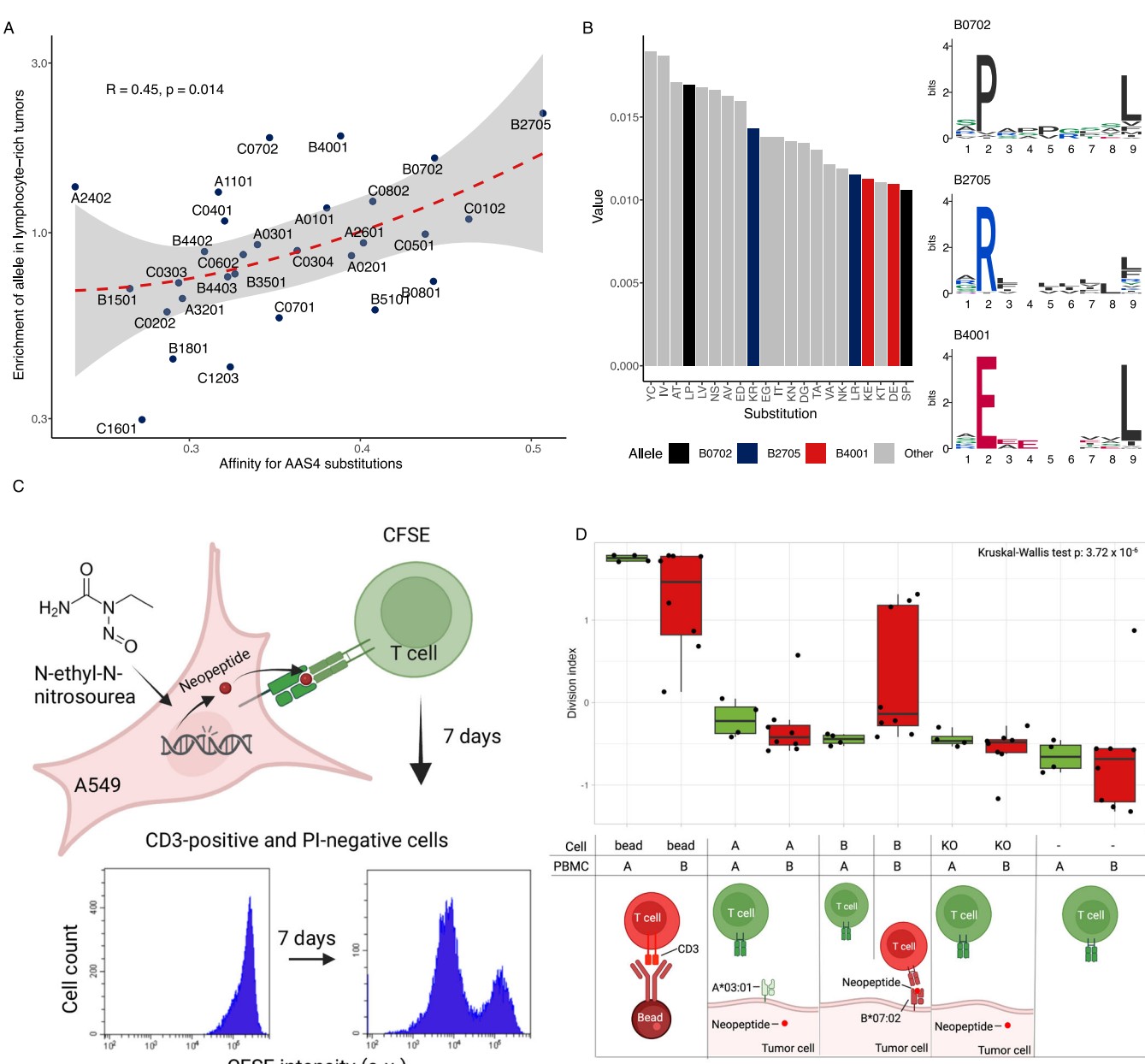

potentially linked to significant neopeptide formation (Appendix Fig. S7 and Appendix Table S1).

To evaluate the immunogenicity of the ENU-treated A549 cells, we co-cultured the ENU-mutagenized A549 cells with peripheral blood mononuclear cells (PBMCs) from healthy individuals (Habib-Agahi et al, 2007) (Fig. 7C). The experimental setup included ENU-treated A549 cells expressing either HLA-A*03:01, HLA-B*07:02, or lacking HLA-I alleles, alongside PBMCs with HLA-A*03:01 (control, $n = 4$), HLA-B*07:02 ($n = 4$), or both ($n = 4$). After a 7-day culture period, we assessed CD3 + T-cell proliferation using the fluorescent cell staining dye, carboxy fluorescein diacetate succinimidyl ester (CFSE), along with propidium iodide (PI) (to detect viable cells) via flow cytometry (Fig. 7C,D, Source Data) (Habib-Agahi et al, 2007; Terrén et al,

2020). This approach provided valuable insights into the ability of ENU-induced neopeptides to stimulate an immune response in specific HLA backgrounds.

In agreement with expectations, we found that the ENU-treated A549 mutant cells expressing HLA-B*07:02 elicited a significantly enhanced immune response, but only when co-cultured with PBMCs from donors who also carried the HLA-B*07:02 allele (Fig. 7D). No other combinations of mutagen treatment and HLA-I genetic background invoked a comparably high proliferation rate of lymphocytes (Fig. 7D). Of note, the HLA-B*07:02-positive cell line accumulated somewhat fewer missense mutations ($n = 1414$) than the HLA-*03:01 positive ($n = 2715$) and the KO ($n = 1653$) cell lines. Therefore, differences in missense mutation numbers are unlikely to explain the link between high lymphocyte proliferation rates and HLA-B*07:02.

◄ **Figure 7. AASs influence antitumor immune response in an HLA-dependent manner.**

(A) HLA alleles that present AAS4-specific substitutions effectively are more likely to be found in lymphocyte-rich samples. The odds ratio of finding a given HLA allele in lymphocyte-rich vs. poor samples is shown for each allele as a function of its specificity to amino acid substitutions found in AAS4-dominant samples. Spearman's rho and the two-sided $P$ value of a correlation test are indicated. (B) HLA alleles commonly found in lymphocyte-rich AAS4-dominant tumours are specific to amino acid substitutions associated with AAS4. The strength of association of amino acid substitutions to AAS4 is shown in decreasing order. The binding logos of the three HLA-B alleles found in lymphocyte-rich tumours are also indicated. (C) This schematic illustrates the experimental approach for assessing T-cell proliferation in response to ENU-treated A549 cells. PBMCs were labelled with the proliferation dye CFSE and co-cultured with tumour cells. The experimental setup included ENU-treated A549 cells expressing either A*03:01, B*07:02, or none (KO), alongside PBMCs carrying HLA-A*03:01, B*07:02, or both. After 7 days of co-culture, CD3 + T-cell proliferation was analysed by flow cytometry. Samples were first gated using anti-CD3 (to identify T cells) and propidium iodide (PI; to exclude dead cells) as described in the "Methods" section. T-cell division was quantified by monitoring CFSE dye dilution: proliferating cells displayed reduced CFSE intensity, forming the G1 peak, while undivided cells remained in the G0 peak. The Division Index, calculated from CFSE profiles (see "Methods"), represents the total number of cell divisions normalised to the initial cell count at the start of the culture, in response to tumour-specific ENU-derived neopeptides. This assay assesses the activation and division of T cells in response to tumour-specific ENU-derived neopeptides. (D) A549 cells carrying HLA-B*07:02 and pre-treated with ENU activate PBMCs of B*07:02-positive donors. The division index of PBMCs after co-culturing them with A549 cells is indicated in different groups. For each PBMC sample (representing a separate biological replicate), division indices were Z-transformed for comparability. A, B, KO cells: A549 cells carrying B*07:02, A*03:01 or no HLA-I alleles, respectively. A (green) and B (red) PBMCs: PMBCs of HLA-A*03:01 ($n = 4$ samples) or B*07:02 ($n = 4$ samples) positive individuals (PBMCs with both A*03:01 and B*07:02 – $n = 4$ samples - were classified as B). The $P$ value of a Kruskal–Wallis test is indicated. FDR-corrected $P$ values of a Dunn's post-hoc test are not indicated due to visualisation purposes; they can be found in Appendix Table S4. On the boxplot, horizontal lines indicate median, boxes indicate the interquartile range (IQR), and whiskers indicate first quartile –1.5 × IQR and third quartile +1.5 × IQR. Source data are available online for this figure.

# Discussion

Exploring mutational signatures has become crucial in cancer genomics, as it can shed light on the environmental and intrinsic elements influencing tumour mutations. Despite significant strides in this area, the link between mutational signatures and cancer immunity remains largely uncharted. While SBS signatures provide a fine-grained description at the nucleotide level, our amino acid substitution signatures (AAS) are intentionally a lower-dimensional, amino acid-centric abstraction designed to capture properties most relevant to antigen presentation and T-cell recognition, rather than to maximise resolution of mutational processes per se. By analysing over 9000 cancer exomes, we identified five distinct classes of AASs, each illustrating specific patterns of amino acid mutations on a genomic scale, arising from both external and internal factors (Table EV1). The primary sources of these five signatures are as follows. AAS1 reflects the effects of tobacco smoke mutagens and various forms of oxidative mutagenesis. AAS2 is triggered by UV light, and DNA damage is repaired by base-excision repair. AAS3 and AAS4 result from different types of defects in the mismatch repair system, with AAS4 is also linked to the mutagenic effects of alkylating agents. Finally, AAS5 is associated with a form of APOBEC-induced mutagenesis.

The analysis revealed specific links between amino acid substitution signatures, environmental mutagens and tumour types (Fig. 3A). For example, AAS1 is prevalent in liver cancer samples and is also associated with aflatoxin exposure. Reassuringly, the link between aflatoxin exposure and liver cancer is well-established (Liu and Wu, 2010; Wu and Santella, 2012), with the mechanisms involving DNA damage, mutagenesis, and promotion of carcinogenesis through oxidative stress and inflammation (Yilmaz et al, 2017). The AAS4 signature predominates in samples from kidney cancer (KIRP, KIRC), and this signature is also linked to glycidamide exposure. Glycidamide is a genotoxic compound metabolised from acrylamide in the human body (Zhivagui et al, 2019). Acrylamide is a chemical compound that forms naturally in certain foods during high-temperature cooking processes, such as frying, baking, or roasting. Several studies have suggested a potential link between acrylamide exposure and the risk of kidney cancer (Zhivagui et al, 2019; Hogervorst et al, 2008; Pelucchi et al, 2015). In addition, oxidative stress induced by acrylamide may contribute to kidney carcinogenesis through mechanisms involving DNA damage and inflammation (Ajibare et al, 2024).

Tumour samples associated with AAS4 are less likely to accumulate hydrophobic amino acids compared to samples linked to other AASs (Fig. 4A), whereas those associated with AAS5 type substitutions tend to accumulate radical amino acid changes, with substantial alterations in charge (Fig. 4B). We proposed that these systematic changes in amino acid physicochemical properties across AASs shape the immunogenic potential of neopeptides generated in these cancer samples. In line with expectation, tumour samples linked to AAS5 displayed an increase in immune-rich tumour microenvironments, while those linked to AAS4 were less likely to display such characteristics (Fig. 5A). Furthermore, tumours associated with AAS4 exhibited characteristics indicative of low immunogenicity, including low lymphocyte infiltration levels, a low M1/M2 macrophage ratio, and reduced numbers of active CD4+ memory cells (Fig. 5B,C). These patterns were observed across various cancer types, regardless of the tumours' mutational burden. In addition, ASS4-dominated tumours were associated with worse prognosis after cancer immunotherapy treatment (Fig. 6A).

The origin of a tumour—including the specific tissue or cell type from which the cancer originates—can significantly influence the tumour's interaction with the immune system (Thorsson et al, 2018). Our research demonstrates that distinct patterns of amino acid substitutions, characteristic of cancers from different origins, have a significant and independent impact on the anticancer immune response. For instance, we observed a high prevalence of a specific amino acid substitution signature (AAS4) in non-UV-associated melanoma samples (Fig. 6B,C). This pattern may contribute to the relatively poor prognosis of patients with non-UV-associated melanomas (Hayward et al, 2017).

We finally explored the relationships between environmental mutagen exposure, HLA variations, and tumour immune characteristics, particularly focusing on why certain rare tumour samples dominated by AAS4 exhibit an effective antitumor response (Fig. 5). We demonstrated that the corresponding patients

possess HLA-I variants capable of efficiently recognising neopeptides with unusual amino acid substitution patterns, such as those prevalent in AAS4-dominated tumours (Fig. 7A,B). One identified protective HLA variant in these patients is HLA-B*07:02, which possesses unique structural features distinguishing it from other variants. HLA-B*07:02 is common in European populations (Gonzalez-Galarza et al, 2019) and is associated with susceptibility to various diseases, including certain cancer types (Kløverpris et al, 2014; Tziotzios et al, 2019; Safaeian et al, 2014). Therefore, understanding the role of HLA-B*07:02 in disease susceptibility and immune responses can guide the development of diagnostic tools tailored to individual genetic profiles.

Our analyses focused primarily on HLA binding and T-cell recognition potential (PRIME score) and did not explicitly model upstream processing steps such as proteasomal cleavage, TAP transport, or aminopeptidase trimming. As a result, our predictions should be interpreted as relative measures of neoantigen quality rather than exact estimates of presentation probability. This limitation is partly mitigated by our validation against immunopeptidomics datasets, which confirm that AAS context influences the likelihood that specific substitutions are observed among HLA-I-bound peptides (Appendix Figs. S16 and S17), but additional processing-aware modelling will be valuable in future work.

Our proof-of-concept experiments demonstrated directly that amino acid substitution signatures can translate into measurable immune responses and highlight links between chemical mutagenesis and tumour immunogenicity. At the same time, the scope of these assays was necessarily limited, as they focused on a single mutagen, tumour model, and specific HLA genotypes only. In the future, broader analyses across additional mutagens, diverse cancer cell lines, and a wider range of HLA alleles will be important to further strengthen the robustness and translational relevance of the AAS framework.

In sum, mutagens and the resulting AASs exhibit significant variability in their ability to induce anticancer immune response, partly in a HLA-I-dependent mechanism.

Our research offers several significant implications for future studies. Firstly, it has been proposed that neopeptide immunogenicity and quality—rather than merely quantity—are critical determinants of cancer immunity and patient survival (McGranahan and Swanton, 2019). We recommend that future research should investigate the combined impact of mutational burden and specific amino acid substitution patterns as biomarkers for predicting responses to immunotherapy. Notably, although intratumoral heterogeneity did not emerge as a confounding factor affecting our results, it remains plausible that the emergence of subclonal populations—particularly under selective pressures such as immune checkpoint blockade—could dynamically reshape the neoantigen landscape and alter the prevalence of AASs over time. Considering this possibility provides additional context for understanding mechanisms of immunotherapy response and resistance. Thus, a deeper exploration of the interplay among mutational burden, substitution patterns, intratumoral dynamics, and immune responses will be essential for developing more effective therapeutic strategies.

Secondly—and related to the previous point—although DNA mismatch repair (MMR) deficiency is typically associated with a high TMB (Chalmers et al, 2017), a large fraction of tumours with MMR deficiency do not show durable responses to immune checkpoint inhibitor therapy (Westcott et al, 2023). This discrepancy suggests that factors beyond neopeptide quantity influence therapeutic outcomes. Our results indicate that MMR-deficient tumours can be characterised by AAS3 and/or AAS4 signatures, with the latter associated with an immune-desert phenotype. These signatures correspond to defects in distinct MMR genes, and their varying prevalence across MMR-deficient tumours may reflect differences in the underlying mechanisms of carcinogenesis. Specifically, MMR-deficient tumours can arise either sporadically or in the context of Lynch syndrome, where a germline MMR mutation is typically followed by a somatic 'second hit'. Notably, mutations in MLH1, MSH2, MSH6, and PMS2, which are the most frequently altered genes in Lynch syndrome (Roht et al, 2023) are closely linked to the AAS4 signature. Taken together, our results, in line with other studies, cast doubt on the strategy of intentionally reducing mismatch repair activity to boost tumour immunogenicity (Westcott et al, 2023; Germano et al, 2017; Helleday, 2019).

Finally, it is important to establish how patterns of small insertions and deletions influence the generation of altered cancer peptides. This is particularly relevant in the context of MMR deficiency, where microsatellite instability frequently generates frameshift mutations and their associated neoantigens. While a systematic evaluation of frameshift-derived immunogenicity would provide valuable insights, it lies beyond the scope of the present study.

In summary, our work underscores the importance of considering both the qualitative aspects of neopeptides and the broader mutational landscape in cancer. By doing so, we can better predict and enhance patient responses to immunotherapy, ultimately improving clinical outcomes.

# Methods

### Reagents and tools table

| Reagent/resource | Reference or source | Identifier or catalogue number |
|---|---|---|
| **Experimental models** | | |
| HLA SUBTYPE PANEL – A549 | Merck | #HLA003-1KT |
| Peripheral blood mononuclear cells (PBMCs) | CTL Europe GmbH | custom lot: CTL_HP1 |
| **Culturing medium components** | | |
| DMEM:F12 medium | Gibco | #21127030 |
| FBS (foetal bovine serum) | Capricorn | #FBS-HI-12A |
| 100x penicillin/ streptomycin solution | Capricorn | #PS-B |
| RPMI medium | Capricorn | #RPMI-A |
| MEM non-essential amino acids | Capricorn | #NEAA-B |
| **S9 mix components** | | |
| Aroclor-1254-induced male Golden Syrian hamster liver | Moltox | #15-03SL.5 |
| NADP | Roche | #10128031001 |

| Reagent/resource | Reference or source | Identifier or catalogue number |
|---|---|---|
| DL-isocitric acid trisodium salt hydrate | Merck | #FLUH9AD3C866 |
| **Mutagenes** | | |
| N-Ethyl-N-nitrosourea | Merck | #N3385-1G |
| Cisplatin | Merck | #P4394-100MG |
| Benz[a]pyrene | Merck | #B1760-250MG |
| **Equipments** | | |
| Solar simulator | Vilber Lourmat | #VL-6.LM |
| CytoFlex S flow cytometer | Beckman Coulter | N/A |
| **Other reagents, kit** | | |
| Trypsin | Capricorn | #TRY-1B |
| PBS (1x) | Capricorn | #PBS-1A |
| gDNA purification kit | Zymo Research | #D3024 |
| carboxyfluorescein succinimidyl ester (CFSE) | Invitrogen | #C34554 |
| Dynabeads™ Human T-Activator CD3/CD28 | Gibco | #11161D |
| interleukin-2 (IL-2) | Invitrogen | #PHC0027 |
| Alexa Fluor® 647 anti-human CD3 antibody | Biolegend | #317312 |
| Propidium iodide (PI) | Merck | #P4170-10MG |
| MycoAlertTM mycoplasma detection kit | Lonza | #LT07-218 |
| **Software** | | |
| R (core environment) | The R Project for Statistical Computing, https://www.r-project.org/ | v4.3.2 |
| Bioconductor (framework) | Huber W et al, Bioinformatics 2005; https://www.bioconductor.org/ | Release 3.18 |
| data.table | Dowle M, Srinivasan A, CRAN; https://cran.r-project.org/package=data.table | v1.14.8 |
| dplyr | Wickham H et al, CRAN; https://cran-r-project.org/package=dplyr | v1.1.4 |
| dunn.test | Dinno A, CRAN; https://CRAN.R-project.org/package=dunn.test | v1.3.6 |
| fastmatch | Eddelbuettel D, CRAN; https://cran.r-project.org/package=fastmatch | v1.1-6 |
| forestmodel | Kennedy N, CRAN; https://CRAN.R-project.org/package=forestmodel | v0.6.2 |

| Reagent/resource | Reference or source | Identifier or catalogue number |
|---|---|---|
| ggplot2 | Wickham H, Springer 2016; https://cran.r-project.org/package=ggplot2 | v3.4.4 |
| ggpubr | Kassambara A, CRAN; https://cran.r-project.org/package=ggpubr | v0.6.0 |
| ggrepel | Slowikowski K, CRAN; https://CRAN.R-project.org/package=ggrepel | v0.9.6 |
| ggseqlogo | Wagih O, CRAN; https://CRAN.R-project.org/package=ggseqlogo | v0.2 |
| ggsimplestats | Heeg M; https://github.com/maximilian-heeg/ggsimplestats | v0.0.1 |
| gridExtra | Auguie B, CRAN; https://cran.r-project.org/package=gridExtra | v2.3 |
| Hmisc | Harrell FE Jr., CRAN; https://cran.r-project.org/package=Hmisc | v5.1-1 |
| kableExtra | Zhu H, CRAN; https://cran.r-project.org/package=kableExtra | v1.3.4 |
| lsa | Wild F, CRAN; https://CRAN.R-project.org/package=lsa | 0.73.3 |
| lubridate | Grolemund G and Wickham H, CRAN; https://cran.r-project.org/package=lubridate | v1.9.4 |
| magick | Ooms J, CRAN; https://CRAN.R-project.org/package=magick | v2.9.0 |
| magrittr | Bache SM and Wickham H, CRAN; https://cran.r-project.org/package=magrittr | v2.0.3 |
| NMF | Gajaux R et al, CRAN; https://cran.r-project.org/package=NMF | v0.28 |
| openxlsx | Walker A, CRAN; https://cran.r-project.org/package=openxlsx | v4.2.5.2 |
| pdftools | Ooms J, CRAN; https://CRAN.R-project.org/package=pdftools | v3.6.0 |
| Peptides | Osorio D et al, CRAN; https://doi.org/10.32614/RJ-2015-001 | v2.4.6 |
| pheatmap | Kolde R, CRAN; https://cran.r-project.org/package=pheatmap | v1.0.12 |
| pixiedust | Nutter B, CRAN; https://CRAN.R-project.org/package=pixiedust | v0.9.4 |
| pROC | Robin X et al, BMC Bioinformatics 2011; https://cran.r-project.org/package=pROC | v1.18.5 |

| Reagent/resource | Reference or source | Identifier or catalogue number |
|---|---|---|
| purrr | Henry L and Wickham H, CRAN; https://cran.r-project.org/package=purrr | v1.0.2 |
| readxl | Wickham H and Bryan J, CRAN; https://cran.r-project.org/package=readxl | v1.4.3 |
| reshape2 | Wickham H, CRAN; https://cran.r-project.org/package=reshape2 | v1.4.4 |
| RISmed | Kovalchik S, CRAN; https://CRAN.R-project.org/package=RISmed | v2.3.0 |
| seqinr | Charif D and Lobry JR, Bioinformatics 2007; https://cran.r-project.org/package=seqinr | v4.2-30 |
| sjPlot | Lüdecke D, CRAN; https://CRAN.R-project.org/package=sjPlot | v2.9.0 |
| stringr | Wickham H, CRAN; https://cran.r-project.org/package=stringr | v1.5.1 |
| survival | Therneau T, CRAN; https://CRAN.R-project.org/package=survival | v3.8-3 |
| survminer | Kassambara A et al, CRAN; https://CRAN.R-project.org/package=survminer | v0.5.1 |
| TCGAbiolinks | Colaprico A et al, Nucleic Acids Res 2016, https://doi.org/10.1093/nar/gkv1507 | v2.38.0 |
| tidyverse | Wickham H et al, J Open Source Softw 2019; https://cran.r-project.org/package=tidyverse | v2.0.0 |
| universalmotif | Tremblay BJM, J Open Source Softw 2019; https://joss.theoj.org/papers/10.21105/joss.07012 | v1.28.0 |
| vcfR | Knaus BJ and Grüner E, Bioinformatics 2017; https://cran.r-project.org/package=vcfR | v1.15.0 |
| CytExpert software | Beckman Coulter | N/A |
| Vactornator | https://www.linearity.io/ | v4.13.5 (20230525122217.032) |

## Cell lines and cell culture

All A549 cell lines were purchased from Merck. Specifically, we obtained the HLA Subtype Panel – A549 (HLA003-1KT, Sigma-Aldrich) and used the following lines: B2M KO (control, HLA003A-1VL; SLCD9117), HLA-A*03:01 (HLA003D-1VL; SLCD9183), and HLA-B*07:02 (HLA003H-1VL; SLCD9187). The identity of cell lines was confirmed by short tandem repeat (STR) profiling by Sigma-Aldrich. A549 offers extensive genomic annotation, has been widely used in mutagenesis (Delhomme

et al, 2023; Muradyan et al, 2011; Zhao et al, 2010; Jiang et al, 2016; Maser et al, 2015) and tumour immunogenicity studies (Javitt et al, 2019; Trojan et al, 2002). Therefore, this cell line is optimal for studying chemical induction of somatic mutations and subsequent analysis of amino acid substitution signatures. The cells were routinely tested for mycoplasma contamination using the MycoAlert™ Mycoplasma Detection Kit (Lonza), and all tests were negative. No blinding was performed; investigators were aware of treatment conditions during cell culture, clonal isolation, sequencing sample submission, and flow cytometry data acquisition/analysis. Cells were cultured in DMEM:F12 (21127030, Gibco) medium supplemented with 10% foetal bovine serum (FBS-HI-12A, Capricorn), penicillin (100 µg/ml), streptomycin (100 U/ml) (PS-B, Capricorn), and maintained at 37 °C in a 5% $CO_2$ incubator.

## Treatment with DNA-damaging agents

The usage of DNA-damaging agents followed established protocols as described by Kucab et al (Kucab et al, 2019). All compounds utilised in this study were dissolved in appropriate solvents and diluted in growth media immediately prior to cell treatment. For compounds requiring metabolic activation, cells were exposed to treatment media containing the S9 mix for 3 h, followed by replacement with fresh growth media. The S9 mix consisted of 0.25% S9 fraction from Aroclor-1254-induced male Golden Syrian hamster liver (#15-03SL.5, Moltox), 3 mM NADP (10128031001, Roche), and 15 mM DL-isocitric acid trisodium salt hydrate (Sigma), prepared in media.

The concentration or dose of each agent that caused 40–60% cytotoxicity was applied as previously described (Kucab et al, 2019). Specific concentrations used in this study included: N-Ethyl-N-nitrosourea (Sigma-Aldrich, N3385-1G) at 400 µM, and Benz[a]pyrene (Sigma-Aldrich, B1760-250MG) at 0.39 µM with S9 mix. Cells were first subcloned and then exposed to the respective compounds or radiation in culture media for up to 24 h. After exposure, growth media were changed and replenished daily as necessary to maintain cell viability and minimise carry-over effects from treatments. For non-chemical agents, cells were exposed to simulated solar radiation (SSR) using a Vilber Lourmat VL-6.LM machine. The SSR output consisted of 10% UVB (295–315 nm) and 90% UVA (315–400 nm), with a total dose of 1.25 J.

## Clonal isolation, genomic DNA extraction and whole-genome sequencing

To isolate single-cell clones, treated cell populations were first dissociated into single-cell suspensions using Trypsin (TRY-1B, Capricorn). The resulting suspensions were seeded at high dilution onto 10 cm petri dishes. Medium was replaced daily, and clones were allowed to establish over a period of 10–12 days. Once established, clones from each treatment condition were sequentially passaged into 24-well plates and subsequently into six-well plates for further expansion. For each clone, cell pellets were collected, and cryopreserved aliquots were prepared. Genomic DNA (gDNA) was isolated from collected cell pellets using a gDNA purification kit (Lonza), following the manufacturer's protocol. The isolated gDNA was then sent to Novogene for sequencing with an average coverage of 30x.

## Peripheral blood mononuclear cell (PBMC) sample preparation

PBMCs were obtained from CTL Europe GmbH (custom lot: CTL_HP1). Cells were washed with 5 ml phosphate-buffered saline (PBS, 1×) and collected by centrifugation at 350 × g for 7 min at room temperature. To remove any residual red blood cells, the collected cell pellet was resuspended in 2 ml ACK lysis buffer (0.15 M NH4Cl, 10 mM KHCO3, 0.1 mM Na2EDTA, pH 7.2–7.4), and incubated for 2 min at room temperature. The suspension was centrifuged at 350 × g for 7 min at room temperature, followed by a wash with 12 ml of complete RPMI medium, and centrifugation under the same conditions.

## T-cell proliferation assay

For the proliferation of T cells, the protocol of Habib-Agahi et al has been adapted and further developed (Habib-Agahi et al, 2007). Cells were stained with carboxyfluorescein succinimidyl ester (CFSE; C34554, Invitrogen) to monitor cell proliferation. After centrifugation, cells were resuspended in 1× PBS at a concentration of $1 \times 10^6$ cells per ml. CFSE was added to the cell suspension at a final concentration of 5 µM, followed by incubation for 10 min at room temperature on a shaker. The staining reaction was halted by washing the cells three times with complete RPMI medium (RPMI-A, Capricorn). Following the washes, PBMCs were resuspended in culture medium composed of RPMI supplemented with 10% foetal bovine serum (FBS), 1% antibiotic–antimycotic, and 1% MEM non-essential amino acids (NEAA-B). To assess cell proliferation, PBMCs were co-cultured with mutagen-treated A549 cells in a 96-well plate. Each well contained 200,000 PBMCs and 4000 A549 cells. For control stimulation, Dynabeads™ Human T-Activator CD3/CD28 (25 µl per $1 \times 10^6$ cells) for T Cell Expansion and Activation (Gibco, 11161D) was included. On the following day, interleukin-2 (IL-2; PHC0027, Invitrogen) was added at a concentration of 1 ng/mL. On day 4 of culture, an additional 50 µl of complete RPMI was added to each well. Cells were incubated and analysed by flow cytometry on the 7th day.

## Immunostaining of PBMCs and flow cytometry analysis

Prior to flow cytometry analysis, PBMCs were collected and stained for T-cell identification and viability assessment. For T-cell staining, 3 µl of Alexa Fluor® 647 anti-human CD3 antibody (BioLegend, 317312) was added to each well, followed by incubation for 45 min at room temperature. Propidium iodide (PI; P4170-10MG, Sigma-Aldrich) was added at a concentration of 10 µg/mL in PBS immediately prior to analysis to exclude dead cells.

Stained cells were analysed using a Beckman Coulter CytoFlex S flow cytometer, with data processed in CytExpert software. Cell division was monitored by tracking CFSE dye dilution, allowing identification of successive generations of dividing cells. PI staining determined cell viability, while anti-CD3 staining specifically identified T cells within the PBMC population.

In the proliferation assay, undivided cells were identified as the G0 peak, characterised by consistent CFSE intensity and coefficient of variation (CV) across samples. Proliferating cells exhibited progressive CFSE dilution, forming the G1 peak. Using flow cytometry gating strategies for PI (to exclude dead cells), anti-CD3 (to identify T cells), and CFSE (to track proliferation), G0 and G1 peaks were clearly distinguished. Using information on G0 and G1 peaks, division index was calculated as follows:

$$DI = \frac{\sum_0^i i \frac{N_i}{2^i}}{\sum_0^i \frac{N_i}{2^i}},$$

where $i$ is the generation number, $N_i$ is the number of cells in generation $i$. Representative flow cytometry images are provided in Source Data.

## Collecting and processing mutational data of TCGA cancer samples

We downloaded data on mutations in 10549 cancer samples of TCGA from the Genomic Data Commons (GDC) portal of the NCBI using the TCGAbiolinks R library (Mounir et al, 2019) (download date: February 27, 2024, all projects, "Single Nucleotide Variation" data category, "Masked Somatic Mutation" data type). In each sample, we kept only missense mutation data for further analysis. We discarded samples with lower than ten missense mutations and determined the fraction of each amino acid substitution in each sample. We included samples with relatively low-mutation counts (<50) since recent evidence indicates that even cancers with modest mutational burdens can generate immunogenic neoantigens (Sethna et al, 2025). To ensure that our findings were not biased by the inclusion of low-mutation samples, we repeated the non-negative matrix factorisation (NMF) using alternative mutation count thresholds of 25 ($n = 7171$ samples) and 50 ($n = 4641$ samples). In both analyses, the resulting amino acid substitution (AAS) profiles showed very high cosine similarity across all substitution signatures (Appendix Table S5). These results confirm that the identified AAS signatures are highly robust and are largely independent of the applied mutation count cutoff.

## Carrying out non-negative matrix factorisation (NMF)

We carried out NMF on the whole dataset using the NMF R library (Gaujoux and Seoighe, 2010). We constructed a matrix including the fraction of each amino acid substitution in every tumour sample. To determine the optimal number of latent factors—corresponding to distinct amino acid substitution signatures—we performed 30 independent runs of the algorithm for each factorisation rank ranging from 2 to 100. Model performance was evaluated using multiple criteria, including cophenetic correlation coefficient, silhouette width, dispersion, explained variance, and the residual sum of squares. A rank of 5 was selected as the most appropriate number of factors. Specifically, the cophenetic correlation remained stable up to rank five but declined thereafter, indicating reduced reproducibility at higher ranks. Similarly, both silhouette width and dispersion metrics supported a five-factor solution, reflecting well-defined clustering and internal coherence. Although the residual sum of squares and explained variance improved marginally with higher ranks, the incremental gains beyond five factors were minimal (Appendix Fig. S14). We evaluated the prevalence of AASs in tumour samples by extracting the basis components from the original amino acid substitution matrix. A sample was defined as being

dominated by a specific AAS if its corresponding basis component exceeded 0.6. In cases where multiple AASs exceeded this threshold (3% of samples), the AAS with the highest value was identified as the dominant one. Samples without any basis component exceeding 0.6 were classified as mixed.

## Simulating the effects of mutagenic processes on amino acid substitutions

We downloaded data on single-base substitution signatures from the COSMIC database (Tate et al, 2019) (https://cancer.sanger.ac.uk/signatures/) and the SIGNAL database (Degasperi et al, 2020) (https://signal.mutationalsignatures.com/). The former contained information on signatures identified in cancer samples, while the latter data were acquired either from mutagenesis or gene knockout experiments. We downloaded the data on the coding genome (Homo_sapiens.GRCh38.cds.all.fa file downloaded from Ensembl FTP site on April 3, 2021) and simulated 1000 random missense mutations for each mutational signature. During simulations, we randomly created single-base substitutions in the coding genome by weighting with the probability of each nucleotide substitution in a given 5' and 3' nucleotide environment found in the signature. We also considered transcriptional strand bias in the case of COSMIC signatures (data on transcriptional strand bias were not available for the SIGNAL database).

## Associating nucleotide mutations with AASs

To determine the association strength between nucleotide mutations in different 5' and 3' environments and AASs, we first randomly created 1000 missense mutations for each of the 192 mutation profiles, which involved the 5' nucleotide ($n = 4$), 3' nucleotide ($n = 4$) and the base substitution ($n = 4*3 = 12$). To each missense mutation, we assigned an AAS randomly, weighting with the association strength between the AAS and the amino acid substitution. We then generated an information count matrix (ICM) motif using the relative fraction of each AAS for each mutation profile. We used the universalmotif R package (Benjamin Jean-Marie, 2024).

## Processing WGS data with IsoMut and Annovar

Raw sequencing data were processed using Novogene's in-house bioinformatics pipeline. Briefly, quality control was carried out to ensure a minimum Q30 value of 80%. The Burrow–Wheeler Aligner (BWA) (Li and Durbin, 2009) v0.7.17 was applied to map reads to the reference human genome (hg38, http://hgdownload.cse.ucsc.edu/goldenPath/hg38/bigZips/analysisSet/hg38.analysisSet.2bit). Sambamba v0.7.1 (Tarasov et al, 2015) was used for sorting the BAM files, and Picard v2.18.9 (Broad Institute, 2019) to mark duplicate reads. Statistics on mapping, coverage and depth for each sample can be found in Appendix Table S6. We detected mutations in cell lines with the IsoMut tool, which is designed to identify experimentally induced mutations in multiple isogenic samples (Pipek et al, 2017). Only samples with at least 10,000 mutations were included in further analysis. Mutations were annotated with Annovar (v. 2020-06-07) (Wang et al, 2010).

## Retrieving PubMed hits for tumours and mutagens

We generated all possible pairs between the full names of tumour types listed in the TCGA and the names of mutagens. Using the RISmed R library, we retrieved the number of PubMed hits containing both the tumour type and mutagen names (Kovalchik, 2021) (search conducted on 03/25/2024). An environmental mutagen was considered as associated with a specific AAS if the cosine similarity between the mutagen-induced substitutions and the given AAS exceeded 0.7. For each AAS-tumour type pair, we calculated the total number of PubMed hits that included the tumour type alongside mutagens linked to the AAS. To estimate the strength of association between AASs and tumour types, for samples belonging to each tumour type, we computed the median of the extracted basis component values separately for AAS1 through AAS5.

## Analysing ICB cohort and treatment-naive melanoma patients

We downloaded mutational and clinical data for ICB-treated patients from ref. (Miao et al, 2018) via cBioPortal (Gao et al, 2013). For each tumour sample, we calculated the fraction of each amino acid substitution. To assess the prevalence of a specific AAS in a sample, we computed the Pearson correlation coefficient between the fractions of amino acid substitutions in the sample and their corresponding association values with the AAS, as defined by the coefficient matrix from NMF. SBS exposures for TCGA samples were obtained from the COSMIC mutational signatures resource (https://cancer.sanger.ac.uk/signatures/). For each melanoma sample, we extracted the fraction of mutations attributed to the UV-related signatures SBS7a–SBS7d and matched these values to the corresponding TCGA barcode. Clinical data for TCGA samples were downloaded from cBioPortal (Gao et al, 2013).

## Validating the effect of AASs on presented neopeptides using immunopeptidomics data

We assembled immunopeptidomics datasets from 17 cancer samples (tumour tissues or cell lines) spanning five tumour types (Appendix Table S7) with publicly available somatic mutation profiles and corresponding mutated peptides. For all possible combinations of sample $s$ and AAS $k$, we computed the cosine similarity $c_{s,k\_}$ between the frequency of the observed amino acid substitutions in sample $s$ and AAS $k$. For each sample, all possible amino acid substitutions ($i = 1,\ldots,166$) were then classified as either detected or not detected in HLA-I–bound neopeptides. Let $a_{i,k}$ denote the association strength of substitution $i$ with AAS $k$. For each substitution $i$ in sample $s$, we defined a weighted sum association strength

$$A_{s,i} = \sum_{k=1}^{5} c_{s,k} a_{i,k}$$

which integrates the contribution of all five AASs according to how similar the sample's substitution profile is to each AAS. Using substitution detection status across samples (detected vs. undetected, $n = 17 \times 166$) as the response variable and $A_{s,i}$ as the predictor, we

constructed receiver operating characteristic (ROC) curves using the pROC R package.

## Determining the amino acid specificity of HLA-I alleles

We obtained a large immunopeptidomic dataset from Sarkizova et al (Sarkizova et al, 2020), which includes data for 92 HLA-I alleles. We focused on alleles with at least 400 reported peptides between 8 and 12 amino acids in length. Peptides of different lengths were analysed separately, and only lengths with at least 100 peptide sequences were analysed for each allele. For each subset of peptides, we calculated the relative frequency of each amino acid at every position. These frequencies were then weighted by the positional importance, following established methods (Henikoff and Henikoff, 1994). Specifically, we calculated the amino acid entropy at each position, reducing the weight of positions with high entropy and emphasising those with more specificity.

The specificity value to each amino acid was then derived by summing the weighted frequencies across all positions, yielding subset-specific values. Finally, we calculated an overall specificity value for each amino acid-allele pair by averaging the subset-specific specificity values, weighted by the relative abundance of the subsets. This provided a comprehensive amino acid specificity value for each HLA allele. The specificity of a given HLA allele to mutated amino acids within AAS1–5 was calculated by first aggregating the coefficient values of each AAS according to the resulting mutated amino acids (i.e., substitution-specific coefficients were summed when they produced the same mutated residue). These aggregated values were then multiplied by the corresponding specificity values of the examined HLA molecules, and the resulting products were summed. For patient-level analyses, the six HLA allele–specific values per individual were averaged.

The information count matrix-type binding logos on Fig. 7B were created with the universalmotif R library using the sequences of 9 amino acid long peptides from ref (Sarkizova et al, 2020).

## Validating amino acid specificity values of HLA-I alleles

Next, we examined whether the calculated AA specificity values indeed predict the presentation of peptides on the cell surface. To this end, we used an independent large-scale immunopeptidomics study (Pearson et al, 2016). In this study, the authors identified HLA-bound peptides of PBMCs collected from healthy individuals. We focused on 9-mer peptides and tested the accuracy of AA specificity values as follows. We pooled all 9-mers found in any samples together.

For each sample, we calculated HLA-specific scores of all 9-mers. For each allele in the given sample, we averaged the position-specific AA-specificity values for each 9-mer peptide. Then, we selected the highest average value corresponding to the strongest binding to HLA in the given sample.

For each sample, we divided the 9-mers to negative peptides (i.e. the ones that were not identified on the surface of the given sample) and positive peptides (identified on the surface of the sample). We created a receiver operating characteristic (ROC) curve for each sample using the calculated scores and considering peptide classification (positive vs. negative peptides). The area under the ROC curve values ranged between 0.73 and 0.87,

suggesting reliable predictive power for HLA amino acid specificity values (Appendix Fig. S15).

## Data availability

The datasets and computer code produced in this study are available in the following databases: WGS data: European Nucleotide Archive PRJEB102539 https://www.ebi.ac.uk/ena/browser/view/PRJEB102539. Computer scripts central to the study: GitHub https://github.com/immunoteam/AAS_publ. R objects, raw datasets for scripts: Zenodo https://doi.org/10.5281/zenodo.17962590.

The source data of this paper are collected in the following database record: biostudies:S-SCDT-10_1038-S44320-026-00193-x.

## Peer review information

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

## Acknowledgements

The authors express their gratitude to Luis Zapata for his insightful suggestions that contributed to the improvement of this manuscript. MM is a partner of the Horizon Health consortium "ID-DarkMatter-NCD" (project number [PN]: 101136582) and gratefully acknowledges support from the EU. The work was supported by the Lendület Programme of the Hungarian Academy of Sciences - grant number LP2025-13/2025; European Union's Horizon 2020 research and innovation program grant No. 739593 (SzJ, BTP, DK, BK and MM); Hungarian Scientific Research Fund grant FK-142312 (MM), FK-131961 (SzJ), K146323 (CP) and PD-146654 (BK); the National Academy of Scientist Education under the sponsorship of the Hungarian Ministry of Innovation and Technology (FEIF/646-4/2021- ITM_SZERZ); KIM NKFIA TKP-2021-EGA-05, KIM NKFIA 2022-2.1.1-NL-2022-00005 (SzJ); NKFI 2020-1.1.6-JÖVŐ-2021-00006 (PB); National Laboratory of Biotechnology Grant 2022-2.1.1-NL-2022-00008 (CP); The European Research Council ERC-2023-ADG 101142626 FutureAntibiotics (CP). MM and BK were supported by the Bolyai János Research Fellowship of the Hungarian Academy of Sciences. The results shown here are in whole or in part based upon data generated by the TCGA Research Network: https://www.cancer.gov/tcga. We are grateful to H, Ye, R Marty, J Font-Burgada, and H Carter for providing the HLA genotype data of TCGA patients. Figures 1, 7C, D, Appendix Fig. S16, and Table EV1 were created with https://BioRender.com.

## Author contributions

**Szilvia Juhász**: Conceptualisation; Formal analysis; Supervision; Funding acquisition; Investigation; Methodology; Writing—original draft; Writing—review and editing. **Benjamin Tamás Papp**: Data curation; Formal analysis; Visualisation. **Anna Tácia Fülöp**: Formal analysis; Validation; Visualisation. **Zoltán Farkas**: Formal analysis; Visualisation; Methodology. **Dávid Kókai**: Formal analysis; Methodology. **Dóra Alexandra Gyémánt**: Data curation; Formal analysis; Investigation. **Franciska Tóth**: Formal analysis; Visualisation. **Zsófia Nacsa**: Investigation. **Dóra Spekhardt**: Investigation. **Balázs Koncz**: Formal analysis; Methodology. **Péter Burkovics**: Conceptualisation; Supervision. **Csaba Pál**: Conceptualisation; Resources; Supervision; Funding acquisition; Writing—original draft; Writing—review and editing. **Máté Manczinger**: Conceptualisation; Resources; Data curation; Software; Formal analysis; Supervision; Funding acquisition; Validation; Investigation; Visualisation; Methodology; Writing—original draft; Project administration; Writing—review and editing.

Source data underlying figure panels in this paper may have individual authorship assigned. Where available, figure panel/source data authorship is listed in the following database record: biostudies:S-SCDT-10_1038-S44320-026-00193-x.

## Funding

## Disclosure and competing interests statement

The authors declare no competing interests.

