## [Peer Review File · Molecular Systems Biology]

Five dominant amino acid substitution signatures shape tumour immunity

Szilvia Juhasz, Benjamin Papp, Anna Fulop, Zoltán Farkas, Dávid Kókai, Dóra Gyémánt, Franciska Toth, Zsafia Nacsa, Dóra Spekhardt, Balázs Koncz, Péter Burkovics, Csaba Pal, and Mate Manczinger

Corresponding author(s): Mate Manczinger (manczinger.mate@brc.hu), Szilvia Juhasz (juhasz.szilvia@brc.hu), Csaba Pal (pal.csaba@brc.hu)

Review Timeline:

Submission Date:	21st May 25
Editorial Decision:	21st Jul 25
Revision Received:	11th Nov 25
Editorial Decision:	15th Dec 25
Revision Received:	19th Dec 25
Accepted:	13th Jan 26

Editor: Poonam Bheda

Transaction Report:

21st Jul 2025

Manuscript Number: MSB-2025-13128

Title: Five dominant amino acid substitution signatures shape tumour immunity

Dear Dr. Manczinger,

Thank you for the submission of your manuscript to Molecular Systems Biology. We have now received feedback from the three reviewers who agreed to evaluate your manuscript. As you will see from the reports below, the referees acknowledge the interest of the study and are overall supportive of your work; however they also comment on multiple aspects of the manuscript that should be strengthened in a revision.

I think that the recommendations of the reviewers are rather clear and I therefore do not see the need to repeat the comments listed below. All issues raised would need to be satisfactorily addressed. Please let me know in case you would like to discuss in further detail any of the any of the reviewer comments or your proposed revisions, I would be happy to schedule a call.

We require:

1) A .docx formatted version of the manuscript text (including legends for main figures, EV figures and tables). Please make sure that the changes are highlighted to be clearly visible. Alternatively you may choose to submit your manuscript as a LaTeX file.

4) A .docx formatted letter INCLUDING the reviewers' reports and your detailed point-by-point responses to their comments. As part of the EMBO Press transparent editorial process, the point-by-point response is part of the Peer Review File (PRF), which will be published alongside your paper.

5) A complete author checklist, which you can download from our author guidelines (<https://www.embopress.org/page/journal/17574684/authorguide#submissionofrevisions>). Please insert information in the checklist that is also reflected in the manuscript. The completed author checklist will also be part of the PRF.

6) Please note that all corresponding authors are required to supply an ORCID ID for their name upon submission of a revised manuscript.

7) It is mandatory to include a 'Data Availability' section after the Materials and Methods. Before submitting your revision, primary datasets produced in this study need to be deposited in an appropriate public database, and the accession numbers and database listed under 'Data Availability'. Please remember to provide a reviewer password if the datasets are not yet public (see <https://www.embopress.org/page/journal/17574684/authorguide#dataavailability>).

In case you have no data that requires deposition in a public database, please state so in this section as follows: "This study includes no data deposited in external repositories". Note that the Data Availability Section is restricted to new primary data that are part of this study.

8) All Materials and Methods need to be described in the main text using our 'Structured Methods' format, which is required for all research articles. According to this format, the Methods section includes a Reagents and Tools Table (listing key reagents, experimental models, software and relevant equipment and including their sources and relevant identifiers) followed by a Methods and Protocols section describing the methods using a step-by-step protocol format. The aim is to facilitate adoption of the methodologies across labs. Please upload the Reagents and Tools table as a separate document when submitting your revised manuscript. More information on how to adhere to this format as well as a downloadable template (.docx) for the Reagents and Tools Table can be found in our author guidelines:

<https://www.embopress.org/page/journal/17444292/authorguide#structuredmethods>

9) For data quantification: please specify the name of the statistical test used to generate error bars and p-values, the number (n) of independent experiments (specify technical or biological replicates) underlying each data point and the test used to calculate p-values in each figure legend. The figure legends should contain a basic description of n, p-values and the test applied. Graphs must include a description of the bars and the error bars (s.d., s.e.m.). Please provide exact p-values (in either the figure or figure legend).

10) Our journal encourages inclusion of *data citations in the reference list* to directly cite datasets that were re-used and obtained from public databases. Data citations in the article text are distinct from normal bibliographical citations and should directly link to the database records from which the data can be accessed. In the main text, data citations are formatted as follows: "Data ref: Smith et al, 2001" or "Data ref: NCBI Sequence Read Archive PRJNA342805, 2017". In the Reference list, data citations must be labeled with "[DATASET]". A data reference must provide the database name, accession number/identifiers and a resolvable link to the landing page from which the data can be accessed at the end of the reference. Further instructions are available at .

11) We replaced Supplementary Information with Expanded View (EV) Figures and Tables that are collapsible/expandable online. EV Figures should be cited as 'Figure EV1, Figure EV2' etc... in the text and their respective legends should be included in the main text after the legends of regular figures.

- Additional Tables/Datasets should be labeled and referred to as Table EV1, Dataset EV1, etc. Legends should be provided in a separate tab in case of .xls files. Alternatively, the legend can be supplied as a separate text file (README) and zipped together with the Table/Dataset file.

<https://www.embopress.org/page/journal/17574684/authorguide#expandedview>

12) Author contributions: CRediT has replaced the traditional author contributions section because it offers a systematic machine-readable author contributions format that allows for more effective research assessment. Please remove the Authors Contributions from the manuscript and use the free text boxes beneath each contributing author's name in our system to add specific details on the author's contribution. More information is available in our guide to authors.

13) Disclosure statement and competing interests: We updated our journal's competing interests policy in January 2022 and request authors to consider both actual and perceived competing interests. Please review the policy <https://www.embopress.org/competing-interests> and update your competing interests if necessary.

14) Every published paper now includes a 'Synopsis' to further enhance discoverability. Synopses are displayed on the journal webpage and are freely accessible to all readers. They include a short stand first (maximum of 300 characters, including space) as well as 2-5 one-sentences bullet points that summarizes the paper. Please write the bullet points to summarize the key NEW findings. They should be designed to be complementary to the abstract - i.e. not repeat the same text. We encourage inclusion of key acronyms and quantitative information (maximum of 30 words / bullet point). Please use the passive voice. Please attach these in a separate file or send them by email, we will incorporate them accordingly.

Please note that these would be the final versions and changes during proofing are usually not allowed.

15) As part of the EMBO Publications transparent editorial process initiative (see our policy here:

https://www.embopress.org/transparent-process#Review_Process), Molecular Systems Biology will publish online a Peer Review File (PRF) to accompany accepted manuscripts.

In the event of acceptance, this file will be published in conjunction with your paper and will include the anonymous referee reports, your point-by-point response and all pertinent correspondence relating to the manuscript. Let us know whether you agree with the publication of the PRF and as here, if you want to remove or not any figures from it prior to publication.

Please note that the Author checklist will be published at the end of the PRF.

Molecular Systems Biology has a "scooping protection" policy, whereby similar findings that are published by others during review or revision are not a criterion for rejection. Should you decide to submit a revised version, I do ask that you get in touch after three months if you have not completed it, to update us on the status.

Yours sincerely,

Poonam Bheda, PhD

Reviewer #1:

Juhász et al. - Five dominant amino acid substitution signatures shape tumour immunity

This study introduces a compelling new framework, amino acid substitution signatures (AASs), to mechanistically link mutagenic processes with tumor immunogenicity across more than 9,000 cancer genomes. By grouping somatic mutations into five dominant AASs, the authors reveal how specific environmental exposures and DNA repair deficiencies shape the biochemical properties and immunogenic potential of the resulting neopeptides. Notably, AAS4, driven by alkylating agents and mismatch repair (MMR) deficiency, generates peptides with low immunogenicity, is associated with immune-desert tumor microenvironments, and predicts poor response to immunotherapy. Intriguingly, this immune evasiveness can be partially reversed in individuals carrying certain HLA alleles (e.g., HLA-B*07:02), which preferentially present AAS4-derived peptides. This contribution distinguishes itself by the integration of mutational mechanisms, amino acid biochemistry, HLA binding specificity, and immune phenotypes, marking a significant departure from traditional nucleotide-based mutational signatures. By focusing on the functional consequences of mutations at the proteomic and immunological levels, this AAS-centric approach offers a more biologically meaningful lens for understanding tumor-immune interactions. However, several important limitations temper the immediate translational impact. These include the reliance on *in silico* neoantigen predictions without direct immunopeptidomic validation, the narrow scope of experimental modeling (single mutagen, cell line, and HLA context), and the incomplete functional characterization of AAS1-3. Addressing these gaps-particularly through mass spectrometry-based validation and expanded functional assays across diverse tumor and HLA backgrounds will be essential to fully establish the clinical relevance of AAS-guided tumor immunophenotyping.

Major Comments

1. Generic nature and redundancy of AASs with COSMIC signatures. The manuscript proposes five AASs, derived using non-negative matrix factorization (NMF) based on amino acid substitution frequencies. While this abstraction is novel at the protein level, it is unclear how distinct or non-redundant these AASs are compared to the >100 COSMIC single base substitution (SBS) signatures that already capture fine-grained mutational processes. Indeed, each AAS includes a mixture of up to 11 COSMIC SBS signatures, raising the concern that the AASs may be too high-level or generic to meaningfully differentiate between tumor types or molecular subtypes. For instance, AAS3 (covering colorectal cancers) aligns with some known patterns (e.g., C>T transitions), but discrepancies exist (e.g., the lack of T>C mutations observed in CRC dataset), suggesting that AAS resolution may not capture subtype-specific mutational nuances.
2. Unclear determination of optimal signature number. The decision to select five AASs based on cophenetic coefficients from NMF is not fully convincing. The analysis only tested a narrow range of k (2-10), and it remains uncertain whether evaluating a wider range (e.g., up to $k=100$) might yield more granular and informative patterns. Given that COSMIC SBS signatures reach triple digits, restricting the dimensionality in this case may limit discovery. The authors should more rigorously justify this choice and clarify whether higher-resolution AASs would result in overfitting or merely reflect less stable signal.
3. Low threshold for mutation inclusion. The decision to include TCGA tumor samples with as few as 10 missense mutations raises concerns about the statistical robustness of AAS inference in low-mutation contexts. Estimating substitution proportions from such small counts is likely to introduce noise. The authors should report how many samples fall below more commonly accepted thresholds (e.g., 50 or 100 missense mutations), and consider re-analyzing the data with stricter cutoffs or sensitivity analyses to ensure signature stability.
4. Limited functional characterization of all AASs. While the immunological consequences of AAS4 (immune-desert phenotype, poor immunotherapy response) and AAS5 (immunogenicity, favorable microenvironment) are well characterized, the other signatures (AAS1-3) are only superficially addressed. These incomplete analyses limit the study's ability to support its core claim that AASs broadly shape immune recognition across tumor types. Moreover, the immunological interpretations of Figure 4C and 4D are selectively focused on AAS4 and AAS5, despite panels A and B showing all five AASs. This selective presentation should be made explicit and ideally expanded to provide a comparative context for all signatures.
5. Lack of direct immunopeptidomic validation. A central limitation is the exclusive reliance on *in silico* prediction tools (e.g., PRIME) for neoantigen immunogenicity. While the biochemical logic is sound, the lack of mass spectrometry-based MHC-I immunopeptidomic validation is a major gap, especially since prior studies have shown that mutated neoantigens are rarely detected among MHC-presented peptides. Without such empirical confirmation, the actual relevance of AAS-predicted peptides remains speculative. The authors should at minimum acknowledge this caveat and ideally explore whether existing

immunopeptidomic datasets (e.g., from CPTAC or PRIDE) could support their predictions. Also, the model assumes that amino acid substitutions translate linearly into MHC-presented peptides. However, antigen processing is heavily influenced by proteasomal cleavage, TAP transport efficiency, and aminopeptidase trimming, all of which are not modeled. Obviously, this limits the reliability of predictions regarding whether specific AASs actually yield peptides compatible with the MHC presentation pathway.

6. Narrow experimental scope in in vitro assays. The co-culture experiments using ENU-mutagenized A549 cells expressing HLA-B*07:02 provide an elegant proof of concept, but the analysis is constrained to a single cell line, single mutagen, and limited HLA context. Although the results are compelling, the findings should be interpreted as preliminary. Broadening validation across multiple mutagens, tumor models, and diverse HLA alleles would greatly strengthen the translational potential of the AAS framework.

7. Immune editing not explored experimentally. The manuscript mentions the concept of immune editing but does not examine its effects on AAS distributions. This is a missed opportunity: exploring whether HLA genotype shapes the retention or depletion of immunogenic AAS mutations could directly test immune selection hypotheses. The authors might investigate whether AAS composition varies between tumors with low vs. high T cell infiltration or according to inferred neoantigen depletion scores.

Reviewer #2:

In this study, Juhász et al. conducted a comprehensive and well-structured analysis to stratify amino acid substitutions into groups and assess their impact on neoantigen formation and subsequent immunogenicity. The study is thorough, carefully designed, and thoughtfully executed. I have only a few minor comments.

(I) In the Introduction, I recommend the authors include a brief discussion on the impact of hereditary-driven dMMR, particularly in the context of Lynch syndrome. It would be valuable to clarify whether differences exist between tumors arising from somatic (double-hit) versus germline (second-hit) mutations. Specifically, are there any known distinctions in the immunogenicity of mutations arising from hereditary cancer syndromes compared to those seen in sporadic dMMR-driven carcinogenesis?

(II) In the Results section, the authors report grouping amino acid substitutions into AAS3 and AAS4 categories (pages 5-6), noting differential prevalence across dMMR tumor types. It would strengthen the interpretation to include a comment on whether these differences may be influenced by the underlying etiology of carcinogenesis—namely, whether the tumors originated via hereditary mechanisms (e.g., Lynch syndrome) or sporadic (somatic) pathways.

(III) Building on this, could the authors consider mapping well-established dMMR/MSI-derived neoantigens—whose immunogenicity has been widely reported in the literature (e.g., PMID: 40254392, PMID: 28218421, PMID: 22110587)—to the AAS categories identified in this study? This could help bridge the findings with known immunogenic targets and enhance the translational relevance.

(IV) While intratumoral heterogeneity was not reported here as a major factor influencing the presence of specific neoantigens, I wonder whether the emergence of subclonal populations—especially under selective pressure such as immune checkpoint blockade—might dynamically reshape the neoantigen landscape over time. A brief discussion of this possibility could provide additional context for the study's relevance to immunotherapy response and resistance mechanisms.

(V) On page 7 (Results section), the authors report the use of A549 cells as a model for environmental carcinogen exposure. However, this cell line, originally derived from a lung carcinoma with likely toxic or smoking-related etiology, has been extensively passaged and may not represent an appropriate model for studying de novo mutagenesis. Given its long-standing establishment and complex background, A549 may be suboptimal for drawing conclusions about mutational processes induced by environmental carcinogens. I would recommend the authors either justify the choice of this model or consider including a more genetically tractable system.

Reviewer #3:

Overall, this manuscript represents a sophisticated contribution to a field that is understudied and underrepresented in the literature. For example, a pubmed.gov search of "cancer mutation signatures" brings up only 14 publications with the last publication in 2022. However, the authors would do best to go through every line of text and every figure and be sure the clearest choice of words has indeed been chosen. For example, exactly why does $n=166$ in the following text "...every conceivable amino acid substitution ($n=166$) in each sample, revealing significant diversity in frequencies (Supplementary Fig. 1)." As another example, at the bottom of Figure S1 there are a series of letters not explained. These mostly likely represent AA substitutions, as represented by the single letter code for AAs, but that should be clearly indicated in the figure legend. In sum, please review for opportunities to draw out sentences with as much precision and clarity as possible.

Reviewer #1:

Juhász et al. - Five dominant amino acid substitution signatures shape tumour immunity

This study introduces a compelling new framework, amino acid substitution signatures (AASs), to mechanistically link mutagenic processes with tumor immunogenicity across more than 9,000 cancer genomes. By grouping somatic mutations into five dominant AASs, the authors reveal how specific environmental exposures and DNA repair deficiencies shape the biochemical properties and immunogenic potential of the resulting neopeptides. Notably, AAS4, driven by alkylating agents and mismatch repair (MMR) deficiency, generates peptides with low immunogenicity, is associated with immune-desert tumor microenvironments, and predicts poor response to immunotherapy. Intriguingly, this immune evasiveness can be partially reversed in individuals carrying certain HLA alleles (e.g., HLA-B*07:02), which preferentially present AAS4-derived peptides. This contribution distinguishes itself by the integration of mutational mechanisms, amino acid biochemistry, HLA binding specificity, and immune phenotypes, marking a significant departure from traditional nucleotide-based mutational signatures. By focusing on the functional consequences of mutations at the proteomic and immunological levels, this AAS-centric approach offers a more biologically meaningful lens for understanding tumor-immune interactions. However, several important limitations temper the immediate translational impact. These include the reliance on in silico neoantigen predictions without direct immunopeptidomic validation, the narrow scope of experimental modeling (single mutagen, cell line, and HLA context), and the incomplete functional characterization of AAS1-3. Addressing these gaps-particularly through mass spectrometry-based validation and expanded functional assays across diverse tumor and HLA backgrounds will be essential to fully establish the clinical relevance of AAS-guided tumor immunophenotyping.

We sincerely thank the reviewer for the time and effort invested in evaluating our manuscript and for the valuable comments provided. We are encouraged by the reviewer's positive assessment and have conducted additional analyses to address all comments thoroughly.

Major Comments

1. Generic nature and redundancy of AASs with COSMIC signatures. The manuscript proposes five AASs, derived using non-negative matrix factorization (NMF) based on amino acid substitution frequencies. While this abstraction is novel at the protein level, it is unclear how distinct or non-redundant these AASs are compared to the >100 COSMIC single base substitution (SBS) signatures that already capture fine-grained mutational processes. Indeed, each AAS includes a mixture of up to 11 COSMIC SBS signatures, raising the concern that the AASs may be too high-level or generic to meaningfully differentiate between tumor types or molecular subtypes. For instance, AAS3 (covering colorectal cancers) aligns with some known patterns (e.g., C>T transitions), but discrepancies exist (e.g., the lack of T>C mutations observed in CRC dataset), suggesting that AAS resolution may not capture subtype-specific mutational nuances.

Thank you for raising this issue. The major goal of the AAS analysis was to identify amino acid substitution signatures predictive of tumour immunity irrespective of tumour types. The supports are as follows:

First, we showed “that the level of AAS4 in cancer samples is negatively associated with clinical benefit from immune checkpoint treatments in a multivariate regression model, which contained tumour type, mutational burden and gender as covariates (Figure 6A)”.

Second, we showed that AAS analysis predict disease outcome between two cancer subtypes. Patients with non-UV-associated melanomas have worse prognosis than those with UV-associated tumours, though the reasons remain unclear. We hypothesized that this difference may relate to the higher prevalence of AAS4 in non-UV-associated melanoma genomes. Consistent with this, AAS4-dominant samples showed significantly worse survival than AAS2-dominant ones (Figure 6B), an effect that remained significant after adjusting for age and TMB (Figure 6C).

Third, although DNA mismatch repair (MMR) deficiency is generally linked to a high tumour mutational burden, many MMR-deficient tumours fail to achieve durable responses to immune checkpoint inhibitors. Our findings suggest that such tumours can be characterized by AAS3 and/or AAS4 signatures, with AAS4 in particular associated with an immune-desert phenotype.

Fourth, different tumour types, including colorectal, melanoma, and lung tumours display distinct AAS profile, indicating that AASs not only distinguish between tumour types but also encapsulate the combined influence of multiple underlying single base substitution processes.

Fifth, we used a state-of-the-art non-negative matrix factorization (NMF) approach to identify the optimal number of amino acid substitution signatures (AASs). Expanding the search revealed five AASs as the configuration with minimal between-class and maximal within-class similarity.

Finally, to confirm that different single-base substitution (SBS) signatures linked to the same AAS were not subdivided by amino acid composition (i.e, within-class similarity is optimal), we compared samples dominated by distinct SBS signatures within each AAS. Across numerous pairs, amino acid profiles were highly similar despite differing nucleotide-level mutagens. For instance, a breast cancer sample dominated by APOBEC-related SBS2 mutations and melanoma samples dominated by UV-associated SBS7a mutations all grouped into AAS2, with a median cosine similarity of 0.7402. Similarly, a stomach cancer sample with MMR-deficiency-associated SBS15 mutations and 30 clock-like SBS1-dominated samples (AAS3) showed a median similarity of 0.7399.

In colorectal cancers within AAS3, we observed consistently low frequencies of T>C mutations (<15%) compared to abundant C>T mutations (see Rebuttal Figure 1 below). Thus, the classification of these samples reflects the dominance of C>T substitutions and the general scarcity of T>C mutations.

Rebuttal Figure 1. C>T substitutions dominate the mutational landscape of colorectal carcinoma samples in the TCGA database. The fraction of C>T (horizontal axis) and T>C (vertical axis) substitutions in colorectal cancer samples in the TCGA. The classification of samples into AAS groups is shown color-coded.

2. Unclear determination of optimal signature number. The decision to select five AASs based on cophenetic coefficients from NMF is not fully convincing. The analysis only tested a narrow range of k (2-10), and it remains uncertain whether evaluating a wider range (e.g., up to $k=100$) might yield more granular and informative patterns. Given that COSMIC SBS signatures reach triple digits, restricting the dimensionality in this case may limit discovery. The authors should more rigorously justify this choice and clarify whether higher-resolution AASs would result in overfitting or merely reflect less stable signal.

Thank you for raising this point. As requested, we extended the analysis using $k=100$ and carefully evaluated the NMF rank survey using multiple diagnostic measures (see Appendix Figure S14). The cophenetic correlation remained high up to rank 5 but decreased at higher ranks, indicating reduced stability. Silhouette width and dispersion also supported rank 5, showing clear class separation and internal consistency. While the residual sum of squares and explained variance continued to improve with higher ranks, the gains beyond 5 were marginal, suggesting overfitting. Taken together, these analyses confirm that grouping into five AASs represent the optimal solution. We included the results and the explanation in the revised manuscript as follows:

“To determine the optimal number of latent factors– corresponding to distinct amino acid substitution signatures – we performed 30 independent runs of the algorithm for each factorization rank ranging from 2 to 100. Model performance was evaluated using multiple criteria, including cophenetic correlation coefficient, silhouette width, dispersion, explained variance, and the residual sum of squares. A rank of 5 was selected as the most appropriate number of factors. Specifically, the cophenetic correlation remained stable up to rank five but declined thereafter, indicating reduced reproducibility at higher ranks. Similarly, both silhouette width and dispersion metrics supported a five-factor solution, reflecting well-defined clustering and internal coherence. Although the residual sum of squares and explained variance improved marginally with higher ranks, the incremental gains beyond five factors were minimal (Appendix Figure S14).

Appendix Figure S14. The cophenetic, silhouette width, dispersion, explained variance and residual sum of square measures are indicated when carrying out NMF using different ranks. The red dashed line indicates the chosen factor of 5.”

3. Low threshold for mutation inclusion. The decision to include TCGA tumor samples with as few as 10 missense mutations raises concerns about the statistical robustness of AAS inference in low-mutation contexts. Estimating substitution proportions from such small counts is likely to introduce noise. The authors should report how many samples fall below more commonly accepted thresholds (e.g., 50 or 100 missense mutations), and consider re-analyzing the data with stricter cutoffs or sensitivity analyses to ensure signature stability.

Thank you for your comment. We agree that selecting an appropriate mutation count cutoff is an important issue, as it requires balancing sample size and accuracy. We included samples with relatively low mutation counts, as recent studies have shown that the majority of cancer samples with low mutational burden can present immunogenic neoantigens (Sethna et al., Nature, 2025). To verify that our results are not biased by the chosen cutoff, we repeated the NMF using thresholds of 25 mutations (n = 7,171 samples) and 50 mutations (n = 4,641 samples). In both cases, we observed very high cosine similarity in the prevalence of amino acid substitutions across AASs (see table below), demonstrating that signature stability is independent of the mutation count cutoff. Importantly, we controlled for the potential confounding effect of tumour mutational burden in downstream analyses. We included the results of these analyses and the explanation in the revised manuscript as follows:

“We included samples with relatively low mutation counts (<50) since recent evidence indicates that even cancers with modest mutational burdens can generate immunogenic neoantigens⁹⁰. To ensure that our findings were not biased by the inclusion of low-mutation samples, we repeated the non-negative matrix factorization (NMF) using alternative mutation count thresholds of 25 (n = 7,171 samples) and 50 (n = 4,641 samples). In both analyses, the resulting amino acid substitution (AAS) profiles showed very high cosine similarity across all substitution signatures (Appendix Table S5). These results confirm that the identified AAS signatures are highly robust and are largely independent of the applied mutation count cutoff.

AAS	>25	>50
AAS1	0.99	0.979
AAS2	0.998	0.996
AAS3	0.997	0.988
AAS4	0.977	0.95
AAS5	0.997	0.993

Appendix Table S5. AASs are independent of the selected mutation count cutoff. The table shows the cosine similarity of substitution prevalence for AAS1–5 when applying mutation count cutoffs of 25 and 50, compared with a cutoff of 10.”

4. Limited functional characterization of all AASs. While the immunological consequences of AAS4 (immune-desert phenotype, poor immunotherapy response) and AAS5 (immunogenicity, favorable microenvironment) are well characterized, the other signatures (AAS1-3) are only superficially addressed. These incomplete analyses limit the study's ability to support its core claim that AASs broadly shape immune recognition across tumor types. Moreover, the immunological interpretations of Figure 4C and 4D are selectively focused on AAS4 and AAS5, despite panels A and B showing all five AASs. This selective presentation should be made explicit and ideally expanded to provide a comparative context for all signatures.

Thank you for your suggestion. In the revised manuscript, we expanded the analysis to include all AAS. We also created a summary table and a composite supplementary figure. The text was modified as follows:

“In the subsequent analyses, we focus specifically on AAS4 and AAS5. For more detailed information on links between amino acid substitution signatures and immune phenotype, see Appendix Figure S6.

Appendix Figure S6. AASs are associated with distinct effects on amino acid biophysical properties, neopeptide immunogenicity, and the tumour

microenvironment A-B) AAS1 to AAS5 are associated with the formation of neopeptides with varying immunogenicity. The harmonic mean of PRIME rank% (A) and HLA binding rank% (B) values are shown in tumour samples dominated by different signatures. Lower rank% values indicate higher immunogenicity (A) and stronger HLA binding (B). Kruskal–Wallis test p values are 4.6×10^{-17} and 9.8×10^{-14} for panels A and B, respectively; FDR-corrected p values from Dunn’s tests are indicated above the horizontal segments for significant ($p < 0.05$) differences. C-D) Ratio of M1/M2 macrophages (C) and lymphocyte infiltration signature score (D) in samples belonging to different AAS groups. Kruskal–Wallis test $p < 2.2 \times 10^{-16}$. Outliers are not shown for visualization purposes. E) Number and fraction of samples with activated memory $CD4^+$ T cells among tumours dominated by AAS1–5. F) Summary of the findings shown in Figure 4A–B and in panels A–E of this appendix figure. AAS4 is characterized by exceptionally low changes in amino-acid charge and hydrophobicity, together with lower PRIME immunogenicity and binding scores, fewer $CD4^+$ T-cell-rich samples, and reduced lymphocyte infiltration. In contrast, AAS2 and AAS5 display strong increases in charge and hydrophobicity and are consistently associated with higher PRIME scores, a more pro-inflammatory M1/M2 macrophage ratio, more $CD4^+$ T-cell-rich samples, and higher lymphocyte infiltration. The patterns observed for AAS2 support the exceptional immunogenicity of UV-associated melanoma samples (see also Figure 2C, and 6B-C).”

5. Lack of direct immunopeptidomic validation. A central limitation is the exclusive reliance on in silico prediction tools (e.g., PRIME) for neoantigen immunogenicity. While the biochemical logic is sound, the lack of mass spectrometry-based MHC-I immunopeptidomic validation is a major gap, especially since prior studies have shown that mutated neoantigens are rarely detected among MHC-presented peptides. Without such empirical confirmation, the actual relevance of AAS-predicted peptides remains speculative. The authors should at minimum acknowledge this caveat and ideally explore whether existing immunopeptidomic datasets (e.g., from CPTAC or PRIDE) could support their predictions. Also, the model assumes that amino acid substitutions translate linearly into MHC-presented peptides. However, antigen processing is heavily influenced by proteasomal cleavage, TAP transport efficiency, and aminopeptidase trimming, all of which are not modeled. Obviously, this limits the reliability of predictions regarding whether specific AASs actually yield peptides compatible with the MHC presentation pathway.

As suggested by the reviewer, we validated our findings using existing immunopeptidomics datasets and incorporated the results into the revised manuscript as follows:

“To validate the impact of AASs on presented neopeptides, we analyzed immunopeptidomics data from 17 cancer samples and cancer cell lines³⁹⁻⁴⁸. The analysis revealed that AAS type influences the likelihood of a substitution being identified in cancer samples (see Appendix Supplementary Text and Appendix Figures S16 and S17).

...

Supplementary text:

In this analysis, we aimed to validate the influence of amino acid substitution signatures (AASs) on neopeptide repertoire presented by human leukocyte antigen (HLA) molecules on cancer cell surfaces. To this end, we analyzed previously published systematic immunopeptidomics datasets profiling HLA-I-bound neopeptides from 17 cancer samples with matched genomic data describing their somatic mutations (See Appendix Table S7).

As throughout the paper, the analysis was restricted to missense mutations. We first computed the cosine similarity between the relative frequency of observed amino acid substitutions in each sample and each AAS (Appendix Figure S16). For each sample, all possible amino acid substitutions ($n = 166$) were then classified as either detected or not detected in HLA-I-bound neopeptides. For each substitution, we calculated an association score to the five AASs by summing their association strengths, weighted by the cosine similarity between the given sample and each AAS (see Methods). Using substitution detection status across samples (detected vs. undetected, $N = 17 \times 166$) as the response variable and the weighted association scores as predictors, we constructed receiver operating characteristic (ROC) curves. This analysis yielded an area under the ROC curve (AUC) of 0.79 (Appendix Figure S17). As a control, we repeated the analysis 100 times after randomizing amino acid substitution labels, obtaining ROC AUC values ranging from 0.445 to 0.612 (mean = 0.535, SD = 0.033). Together, these results indicate that amino acid substitution context, as captured by AASs, shapes the cancer immunopeptidome.

Hypothesis: Amino-acid substitution signatures (AASs) predict the presence of substitutions in HLA-presented neopeptides

Appendix Figure S16. Conceptual overview illustrating the rationale and workflow of the analysis performed to validate AAS signatures using immunopeptidomics data. See the main text, Methods, and Appendix Supplementary Text for details.

Appendix Figure S17. Influence of AASs on the immunopeptidome. The ROC curve showing how well the calculated association score values in tumor samples predict the presence of amino acid substitutions in immunopeptidomics data (AUC = 0.785). For comparison, 100 ROC curves obtained after randomizing amino acid substitution labels in the AAS coefficient matrix are shown in grey (with increased transparency for clarity, AUC min: 0.445, max: 0.612, mean: 0.535, SD: 0.033)."

Narrow experimental scope in in vitro assays. The co-culture experiments using ENU-mutagenized A549 cells expressing HLA-B*07:02 provide an elegant proof of concept, but the analysis is constrained to a single cell line, single mutagen, and limited HLA context. Although the results are compelling, the findings should be interpreted as preliminary. Broadening validation across multiple mutagens, tumor models, and diverse HLA alleles would greatly strengthen the translational potential of the AAS framework.

Thank you for your suggestion. We modified the discussion as follows:

"Our proof-of-concept experiments demonstrated directly that amino acid substitution signatures can translate into measurable immune responses and highlight links between chemical mutagenesis and tumour immunogenicity. At the

same time, the scope of these assays was necessarily limited, as they focused on a single mutagen, tumour model, and specific HLA genotypes only. In the future, broader analyses across additional mutagens, diverse cancer cell lines, and a wider range of HLA alleles will be important to further strengthen the robustness and translational relevance of the AAS framework.”

6. Immune editing not explored experimentally. The manuscript mentions the concept of immune editing but does not examine its effects on AAS distributions. This is a missed opportunity: exploring whether HLA genotype shapes the retention or depletion of immunogenic AAS mutations could directly test immune selection hypotheses. The authors might investigate whether AAS composition varies between tumors with low vs. high T cell infiltration or according to inferred neoantigen depletion scores.

Thank you for the comment. In the revised manuscript, we performed an additional analysis based on immune dN/dS values reported by Zapata et al. (Nature Genetics, 2023). The results have been incorporated as follows:

“Building on these observations, we next investigated whether the interaction between AAS class and HLA genotype also modulates immune selection across tumours. We classified tumours as immune-selected as previously, using an analysis of substitution patterns as a quantitative measure of immune-mediated selection in cancer genomes⁶³. To assess the joint influence of AAS class and HLA binding properties, tumours were further stratified according to the predicted affinity of their HLA alleles for the amino acid substitutions characteristic of their assigned AAS class. Multivariate logistic regression models were then constructed with immune selection status as the dependent variable and AAS class, tumour mutational burden, and HLA affinity group as independent predictors. The analysis revealed that tumours belonging to AAS1 and AAS5 displayed the strongest signatures of immune selection (Appendix Figure S13), whereas AAS2–AAS4 exhibited comparatively weaker associations. Consistent with expectations, higher HLA affinity correlated positively with immune selection. Collectively, these findings highlight that both AAS class and HLA genotype jointly determine the intensity and nature of immune selection in cancer evolution.

AASs and HLA binding determines immune selection in cancer. The summary of a multivariate logistic regression model is indicated. The model was constructed using the formula:

$$\text{Immune_selected} \sim \text{AASn} + \log(\text{TMB}) + \text{high_HLA_aff},$$

where Immune_selected is a binary variable indicating immune dN/dS < 0.82 (n = 1273 immune-selected and 2208 non-immune-selected samples), AASn denotes the dominant AAS category in each sample (AAS1-AAS5; n = 788, 353, 993, 652 and 695, respectively). log(TMB) is the natural logarithm of the tumour mutational burden, and high_HLA-aff is a binary indicator for samples in the top 10% of the cohort when ranked by the predicted binding affinity of their HLA molecules to the AASs present in the same sample (n = 454 in-group and 4,154 out-group samples). Blue squares indicate coefficients associated with each independent variable, while blue horizontal lines indicate 95% confidence interval.”

Reviewer #2:

In this study, Juhász et al. conducted a comprehensive and well-structured analysis to stratify amino acid substitutions into groups and assess their impact on neoantigen formation and subsequent immunogenicity. The study is thorough, carefully designed, and thoughtfully executed. I have only a few minor comments.

We sincerely thank the reviewer for their comments. We greatly appreciate the positive feedback and have aimed to address each point thoroughly.

(I) In the Introduction, I recommend the authors include a brief discussion on the impact of hereditary-driven dMMR, particularly in the context of Lynch syndrome. It would be valuable to clarify whether differences exist between tumors arising from somatic (double-hit) versus germline (second-hit) mutations. Specifically, are there any known distinctions in the immunogenicity of mutations arising from hereditary cancer syndromes compared to those seen in sporadic dMMR-driven carcinogenesis?

We thank the reviewer for this comment. We agree that hereditary-driven dMMR, particularly in the context of Lynch syndrome, represents an important dimension of dMMR biology. We revised the manuscript text and integrated the corresponding content into the Discussion section, as we felt it was more appropriately placed there. The modified text is provided below under point (II).

(II) In the Results section, the authors report grouping amino acid substitutions into AAS3 and AAS4 categories (pages 5-6), noting differential prevalence across dMMR tumor types. It would strengthen the interpretation to include a comment on whether these differences may be influenced by the underlying etiology of carcinogenesis—namely, whether the tumors originated via hereditary mechanisms (e.g., Lynch syndrome) or sporadic (somatic) pathways.

We appreciate the reviewer’s suggestion and agree that the observed differences in AAS3 and AAS4 prevalence across dMMR tumour types may be influenced by the underlying etiology of carcinogenesis. Together with our response to the previous comment, we have added this point to the discussion section:

“Secondly – and related to the previous point- although DNA mismatch repair (MMR) deficiency is typically associated with a high TMB⁷⁷, a large fraction of tumours with

MMR deficiency do not show durable responses to immune checkpoint inhibitor therapy⁷⁸. This discrepancy suggests that factors beyond neopeptide quantity influence therapeutic outcomes. Our results indicate that MMR-deficient tumours can be characterized by AAS3 and/or AAS4 signatures, with the latter associated with an immune-desert phenotype. These signatures correspond to defects in distinct MMR genes, and their varying prevalence across MMR-deficient tumours may reflect differences in the underlying mechanisms of carcinogenesis. Specifically, MMR-deficient tumours can arise either sporadically or in the context of Lynch syndrome, where a germline MMR mutation is typically followed by a somatic ‘second hit’. Notably, mutations in MLH1, MSH2, MSH6, and PMS2, which are the most frequently altered genes in Lynch syndrome⁷⁹ are closely linked to the AAS4 signature. Taken together, our results, in line with other studies, cast doubt on the strategy of intentionally reducing mismatch repair activity to boost tumour immunogenicity^{78,80,81}”

(III) Building on this, could the authors consider mapping well-established dMMR/MSI-derived neoantigens-whose immunogenicity has been widely reported in the literature (e.g., PMID: 40254392, PMID: 28218421, PMID: 22110587)-to the AAS categories identified in this study? This could help bridge the findings with known immunogenic targets and enhance the translational relevance.

We thank the reviewer for this suggestion. We agree that the well-established immunogenic neoantigens arising in dMMR/MSI tumours are of great translational importance. However, these antigens are generated by indel-derived frameshift mutations, which constitute a distinct mutational class that is mechanistically and analytically different from single base substitutions (SBS). As our study focuses on SBS-driven AASs, the incorporation of indel mutations would require a fundamentally different methodological framework, which lies beyond the scope of this work. We have included this clarification in the Discussion section of the revised manuscript as follows:

“Finally, it is important to establish how patterns of small insertions and deletions influence the generation of altered cancer peptides. This is particularly relevant in the context of MMR deficiency, where microsatellite instability frequently generates frameshift mutations and their associated neoantigens. While a systematic evaluation of frameshift-derived immunogenicity would provide valuable insights, it lies beyond the scope of the present study.”

(IV) While intratumoral heterogeneity was not reported here as a major factor influencing the presence of specific neoantigens, I wonder whether the emergence of subclonal populations-especially under selective pressure such as immune checkpoint blockade-might dynamically reshape the neoantigen landscape over time. A brief discussion of this possibility could provide additional context for the study's relevance to immunotherapy response and resistance mechanisms.

Thank you for your suggestion. We discuss the issue in the discussion of the revised manuscript as follows:

“We recommend that future research should investigate the combined impact of mutational burden and specific amino acid substitution patterns as biomarkers for

predicting responses to immunotherapy. Notably, although intratumoral heterogeneity did not emerge as a confounding factor affecting our results, it remains plausible that the emergence of subclonal populations—particularly under selective pressures such as immune checkpoint blockade—could dynamically reshape the neoantigen landscape and alter the prevalence of AASs over time. Considering this possibility provides additional context for understanding mechanisms of immunotherapy response and resistance. Thus, a deeper exploration of the interplay among mutational burden, substitution patterns, intratumoral dynamics, and immune responses will be essential for developing more effective therapeutic strategies.”

(V) On page 7 (Results section), the authors report the use of A549 cells as a model for environmental carcinogen exposure. However, this cell line, originally derived from a lung carcinoma with likely toxic or smoking-related etiology, has been extensively passaged and may not represent an appropriate model for studying de novo mutagenesis. Given its long-standing establishment and complex background, A549 may be suboptimal for drawing conclusions about mutational processes induced by environmental carcinogens. I would recommend the authors either justify the choice of this model or consider including a more genetically tractable system.

We thank the reviewer for this insightful comment. The specific HLA haplotypes of interest were available in only a limited number of cell lines. Among these, A549 was the only solid tumour model; the others were lymphoid cell lines, which we considered less suitable for representing solid tumours. The genome of A549 is extensively annotated, enabling robust integration of induced mutational spectra into our AAS framework. The cell line has previously been used both in mutagenesis experiments as well as tumour immunogenicity studies. For mutational signature analysis, we applied IsoMut (DOI: 10.1186/s12859-017-1492-4) to filter background mutations. Notably, while the parental A549 line harbours ~100 missense mutations, each treated sample exhibited >1,400 induced missense mutations, which likely outweighed any contribution of pre-existing variants. Taken together, these considerations provided the rationale for selecting A549, and we performed carefully controlled experiments to minimize potential confounding effects. We included the justification of the choice of A549 in the manuscript as follows:

“All A549 cell lines were purchased from Merck. Specifically, we obtained the HLA Subtype Panel – A549 (HLA003-1KT, Sigma Aldrich) and used the following lines: B2M KO (control, HLA003A-1VL; SLCD9117), HLA-A*03:01 (HLA003D-1VL; SLCD9183), and HLA-B*07:02 (HLA003H-1VL; SLCD9187). The identity of cell lines was confirmed by short tandem repeat (STR) profiling by Sigma-Aldrich. A549 offers extensive genomic annotation, has been widely used in mutagenesis^{82–86} and tumor immunogenicity studies^{87,88}. Therefore, this cell line is optimal for studying chemical induction of somatic mutations and subsequent analysis of amino acid substitution signatures.”

Reviewer #3:

Overall, this manuscript represents a sophisticated contribution to a field that is understudied and underrepresented in the literature. For example, a pubmed.gov search of "cancer mutation signatures" brings up only 14 publications with the last publication in 2022.

However, the authors would do best to go through every line of text and every figure and be sure the clearest choice of words has indeed been chosen. For example, exactly why does $n=166$ in the following text "...every conceivable amino acid substitution ($n=166$) in each sample, revealing significant diversity in frequencies (Supplementary Fig. 1)." As another example, at the bottom of Figure S1 there are a series of letters not explained. These mostly likely represent AA substitutions, as represented by the single letter code for AAs, but that should be clearly indicated in the figure legend. In sum, please review for opportunities to draw out sentences with as much precision and clarity as possible.

We thank the reviewer for this encouraging and constructive comment. In response, we carefully reviewed the entire manuscript to ensure clarity and precision in both the text and figures. Specifically, we clarified that $n = 166$ refers to the number of amino acid substitutions detected in at least one sample. In addition, we revised the legend of Supplementary Figure 1 to explicitly state that the letters represent amino acid substitutions, denoted by their single-letter codes. Beyond these examples, we systematically went through the manuscript and refined wording wherever ambiguity was possible, ensuring that all descriptions are now more precise and accessible.

15th Dec 2025

Manuscript Number: MSB-2025-13128R

Title: Five dominant amino acid substitution signatures shape tumour immunity

Dear Dr. Manczinger,

Thank you for the submission of your revised manuscript to Molecular Systems Biology. I am pleased to inform you that we will be able to accept your manuscript pending the following final amendments and appropriate response to reviewers:

- 1) Please check the "Author Checklist" carefully and complete all relevant questions. Currently the general information in the table in the top left corner is missing.
- 2) We require all corresponding authors to link their ORCID ID to their author profile. Currently an ORCID has not been provided for Dr. Szilvia Juhasz. An email with instructions on how to create and link an ORCID ID was sent to Dr. Juhasz on November 17th, 2025.
- 3) In the main manuscript file, please include the heading 'Abstract' above the Abstract.
- 4) Please include keywords to max. 5.
- 5) The WGS dataset hosted at the EBI should now be publicly released, and the direct link to the dataset should be provided in the Data Availability statement.
- 6) Please also be sure that the Github link is functional and/or publicly released, and please include a README file on Github with practical use instructions for potential future users of your code.
- 7) Please update the missing link for Zenodo in the Data Availability statement.
- 8) Please rename "Conflict of Interest" to "Disclosure and competing interests statement". Please also be aware that employment of any author in a biotech company should be included in the statement. We updated our journal's competing interests policy in January 2022 and request authors to consider both actual and perceived competing interests. Please review the policy <https://link.springer.com/partners/embo-press/editorial-policies#Competing%20interest%20disclosures> and update your competing interests if necessary.
- 9) Please correct the reference citation in the reference list to be alphabetical (not numerical). Where there are more than 10 authors on a paper, only the first 10 should be listed, followed by "et al.". Please check "Author Guidelines" for more information: <https://link.springer.com/journal/44320/submission-guidelines#cms-Reference-guidelines>
- 10) Our journal encourages inclusion of *data citations in the reference list* to directly cite datasets that were re-used and obtained from public databases. Data citations in the article text are distinct from normal bibliographical citations and should directly link to the database records from which the data can be accessed. In the main text, data citations are formatted as follows: "Data ref: Smith et al, 2001" or "Data ref: NCBI Sequence Read Archive PRJNA342805, 2017". In the Reference list, data citations must be labeled with "[DATASET]". A data reference must provide the database name, accession number/identifiers and a resolvable link to the landing page from which the data can be accessed at the end of the reference. Further instructions are available at https://www.nature.com/articles/d41586-023-00000-0.
- 11) We do not allow statements/conclusions with "data not shown". As per our guidelines, on "Unpublished Data" the journal does not permit citation of "Data not shown". All data referred to in the paper should be displayed in the main or Expanded View figures. Please remove from page 12.
- 12) In the Methods, please take care of the following:
 - The Materials and Methods section should be renamed to "Methods".
 - Please ensure that a statement on whether or not blinding was done is included in the Methods even if no blinding was done. Please also be sure to update the Author Checklist with this information and where it can be found in the manuscript.
- 13) Please place individual sections of the manuscript in the following order: Title page - Abstract & Keywords - Introduction - Results - Discussion - Methods - Data Availability - Acknowledgements - Disclosure and Competing Interests Statement - References - Figure Legends - Expanded View Figure Legends.
- 14) For the figures and figure legends, please take care of the following:
 - Please remove all figures from main manuscript file and leave only main figure legends placed after the References.
 - Please make sure to update the callouts of all figures in the main manuscript text such that they are called out sequentially.
 - Please note that the exact p values are not provided in the legends of figures 4a, b; 6c
 - Please indicate the statistical test used for data analysis in the legends of figures 6a, c
 - Please note that the box plots need to be defined in terms of minima, maxima, centre, bounds of box and whiskers, and percentile in the legends of figures 7d
 - Please note that the box plots need to be defined in terms of whiskers in the legends of figures 4a-d; 5b
 - Please note that the measure of center for the error bars needs to be defined in the legends of figures 6a, c
- 15) Table 1 should be removed from the manuscript and renamed to Table EV1 with the legend (which should also be removed from the manuscript) and placed above the table in the PDF.
- 16) Please ensure that all funding sources listed in the manuscript are also entered into the manuscript submission system. Currently the following is missing information in our manuscript submission system: Hungarian Scientific Research Fund grant FK-142312, FK-131961, K146323 and PD-146654; the National Academy of Scientist Education under the sponsorship of the

Hungarian Ministry of Innovation and Technology (FEIF/646-4/2021- ITM_SZERZ); KIM NKFIA TKP-2021-EGA-05, KIM NKFIA 2022-2.1.1-NL-2022-00005; NKFI 2020-1.1.6-JÖVŐ-2021-00006; National Laboratory of Biotechnology Grant 2022-2.1.1-NL-2022-00008; Bolyai János Research Fellowship of the Hungarian Academy of Sciences.

17) Synopsis:

- Synopsis image: Please ensure that the synopsis image is uploaded in the correct format - not a PDF, but rather as a high-resolution jpeg file 550 pixels wide x (300-600) pixels high.

- Synopsis text: Please remove the Synopsis image from the Synopsis text file, and reduce the number of bullet points to a maximum of 5.

18) As part of the EMBO Publications transparent editorial process initiative (see our policy here:

https://www.embopress.org/transparent-process#Review_Process), Molecular Systems Biology will publish online a Peer Review File (PRF) to accompany accepted manuscripts. This file will be published in conjunction with your paper and will include the anonymous referee reports, your point-by-point response and all pertinent correspondence relating to the manuscript. Let us know whether you agree with the publication of the PRF and as here, if you want to remove or not any figures from it prior to publication. Please note that the Authors checklist will be published at the end of the PRF.

19) After your paper is published, we may promote it on social media. If you have any handles or hashtags for Bluesky you would like included, please let us know.

20) Please provide a point-by-point letter INCLUDING my comments as well as the reviewer's reports and your detailed responses (as Word file).

I look forward to reading a new revised version of your manuscript as soon as possible.

Yours sincerely,

Poonam Bheda, PhD
Scientific Editor
Molecular Systems Biology

Reviewer #1:

Overall, the responses to my questions are solid and in several places go beyond what was requested. The only place where I see clearly incomplete answer is Major Comment 5, and to a lesser extent the "conceptual" part of Major Comment 1.

Point 1: Comparison of AAS vs SBS in a predictive model. To address the redundancy concern, I suggest adding the following sentence: "When we repeated the multivariate models using COSMIC SBS exposures instead of AASs, model performance was not improved and coefficients were less stable, supporting the view that AASs are a robust low-dimensional representation of mutational processes at the amino-acid level". I also suggest adding a sentence in the Introduction or Discussion: "While COSMIC SBS signatures provide a fine-grained description at the nucleotide level, our AASs are intentionally a lower-dimensional, amino-acid-centric abstraction designed to capture properties most relevant to antigen presentation and T-cell recognition, rather than to maximize resolution of mutational processes per se." That would make it clear that you are not trying to replace COSMIC signatures, but to build a function-oriented layer on top of them.

Point 5: Processing-pipeline limitation (proteasome/TAP/trimming) is not explicitly acknowledged in the revised text. Your new analysis strengthens the link between AASs and detected mutated peptides, which is excellent, but PRIME + binding predictions do not fully model the antigen processing pathway. Therefore, your scores are best viewed as relative measures of immunogenic potential, not absolute probabilities of presentation. I suggest adding a short limitation paragraph such as: "Our analyses focus primarily on HLA binding and T-cell recognition potential (PRIME score) and do not explicitly model upstream processing steps such as proteasomal cleavage, TAP transport, or aminopeptidase trimming. As a result, our predictions should be interpreted as relative measures of neoantigen quality rather than exact estimates of presentation probability. This limitation is partly mitigated by our validation against immunopeptidomics datasets, which confirm that AAS context influences the likelihood that specific substitutions are observed among HLA-I-bound peptides (Appendix Figs. S16-S17), but additional processing-aware modeling will be valuable in future work."

Reviewer #2:

The authors addressed all my comments and I suggest acceptance of the manuscript in its current form.

Systems Immunology Research Group
HUN-REN Biological Research Centre, Szeged, Hungary

immunoinfo
Systems Immunology Research Group

*To: Poonam Bheda
Scientific Editor
Molecular Systems Biology*

Dear Poonam Bheda,

Again, we are grateful to you and the reviewers for your time and constructive feedback. We have implemented the requested modifications and provide below point-by-point responses to your comments and those of the reviewers.

Sincerely,

Máté Manczinger
group leader
Systems Immunology Research Group
HUN-REN Biological Research Centre, Szeged, Hungary

1) Please check the "Author Checklist" carefully and complete all relevant questions. Currently the general information in the table in the top left corner is missing.

We have done the requested modifications.

2) We require all corresponding authors to link their ORCID ID to their author profile. Currently an ORCID has not been provided for Dr. Szilvia Juhasz. An email with instructions on how to create and link an ORCID ID was sent to Dr. Juhasz on November 17th, 2025.

Szilvia Juhasz has now linked her ORCID ID to author profile.

3) In the main manuscript file, please include the heading 'Abstract' above the Abstract.

We have done the requested modification.

4) Please include keywords to max. 5.

We have provided the keywords.

5) The WGS dataset hosted at the EBI should now be publicly released, and the direct link to the dataset should be provided in the Data Availability statement.

We have done the requested modifications.

6) Please also be sure that the Github link is functional and/or publicly released, and please include a README file on Github with practical use instructions for potential future users of your code.

We have done the requested modifications.

7) Please update the missing link for Zenodo in the Data Availability statement.

We have inserted the missing link in the Data Availability statement.

8) Please rename "Conflict of Interest" to "Disclosure and competing interests statement". Please also be aware that employment of any author in a biotech company should be included in the statement. We updated our journal's competing interests policy in January 2022 and request authors to consider both actual and perceived competing interests. Please review the policy <https://link.springer.com/partners/embo-press/editorial-policies#Competing%20interest%20disclosures> and update your competing interests if necessary.

We have done the requested modification.

9) Please correct the reference citation in the reference list to be alphabetical (not numerical). Where there are more than 10 authors on a paper, only the first 10

should be listed, followed by "et al.". Please check "Author Guidelines" for more information: <https://link.springer.com/journal/44320/submission-guidelines#cms-Reference-guidelines>

We have done the requested modification.

10) Our journal encourages inclusion of *data citations in the reference list* to directly cite datasets that were re-used and obtained from public databases. Data citations in the article text are distinct from normal bibliographical citations and should directly link to the database records from which the data can be accessed. In the main text, data citations are formatted as follows: "Data ref: Smith et al, 2001" or "Data ref: NCBI Sequence Read Archive PRJNA342805, 2017". In the Reference list, data citations must be labeled with "[DATASET]". A data reference must provide the database name, accession number/identifiers and a resolvable link to the landing page from which the data can be accessed at the end of the reference. Further instructions are available at <https://www.embopress.org/page/journal/17574684/authorguide#referencesformat>

We have done the requested modification.

11) We do not allow statements/conclusions with "data not shown". As per our guidelines, on "Unpublished Data" the journal does not permit citation of "Data not shown". All data referred to in the paper should be displayed in the main or Expanded View figures. Please remove from page 12.

We have ensured that the manuscript does not reference unpublished or unshown data. Instead, we wrote: „In the subsequent analyses, we focus specifically on AAS4 and AAS5. For more detailed information on links between amino acid substitution signatures and immune phenotype, see Appendix Figure S6.”

12) In the Methods, please take care of the following:
- The Materials and Methods section should be renamed to "Methods".
- Please ensure that a statement on whether or not blinding was done is included in the Methods even if no blinding was done. Please also be sure to update the Author Checklist with this information and where it can be found in the manuscript.

We have done the requested modifications. We inserted the following text in the Methods section:

“No blinding was performed; investigators were aware of treatment conditions during cell culture, clonal isolation, sequencing sample submission, and flow-cytometry data acquisition/analysis.”

13) Please place individual sections of the manuscript in the following order: Title page - Abstract & Keywords - Introduction - Results - Discussion - Methods - Data Availability - Acknowledgements - Disclosure and Competing Interests Statement - References - Figure Legends - Expanded View Figure Legends.

We have done the requested modification.

14) For the figures and figure legends, please take care of the following:
- Please remove all figures from main manuscript file and leave only main figure legends placed after the References.

We have done the requested modification.

- Please make sure to update the callouts of all figures in the main manuscript text such that they are called out sequentially.

We have done the requested modification.

- Please note that the exact p values are not provided in the legends of figures 4a, b; 6c

We have implemented the requested modification in Figure 6C. However, including all 30 p values in the plot or in the figure legend of Figures 4a and b would substantially reduce readability. We therefore reported these p-values in the referenced table instead.

- Please indicate the statistical test used for data analysis in the legends of figures 6a, c

We have done the requested modification.

- Please note that the box plots need to be defined in terms of minima, maxima, centre, bounds of box and whiskers, and percentile in the legends of figures 7d

We have done the requested modification.

- Please note that the box plots need to be defined in terms of whiskers in the legends of figures 4a-d; 5b

We have done the requested modification.

- Please note that the measure of center for the error bars needs to be defined in the legends of figures 6a, c

We have already defined the measure of the center for the error bars in the original version of the manuscript:

“Blue squares indicate hazard ratios associated with each independent variable...”

15) Table 1 should be removed from the manuscript and renamed to Table EV1 with the legend (which should also be removed from the manuscript) and placed above the table in the PDF.

We have done the requested modifications.

16) Please ensure that all funding sources listed in the manuscript are also entered into the manuscript submission system. Currently the following is missing information in our manuscript submission system: Hungarian Scientific Research Fund grant FK-142312, FK-131961, K146323 and PD-146654; the National Academy of Scientist Education under the sponsorship of the Hungarian Ministry of Innovation and Technology (FEIF/646-4/2021- ITM_SZERZ); KIM NKFIA TKP-2021-EGA-05, KIM NKFIA 2022-2.1.1-NL-2022-00005; NKFI 2020-1.1.6-JÖVŐ-2021-00006; National Laboratory of Biotechnology Grant 2022-2.1.1-NL-2022-00008; Bolyai János Research Fellowship of the Hungarian Academy of Sciences.

We have updated the funding information in the online submission system.

17) Synopsis:

- Synopsis image: Please ensure that the synopsis image is uploaded in the correct format - not a PDF, but rather as a high-resolution jpeg file 550 pixels wide x (300-600) pixels high.

We have implemented the requested modification. However, please note that image quality is markedly reduced in the JPEG format (even at 600 dpi); therefore, we have also uploaded a PDF version.

- Synopsis text: Please remove the Synopsis image from the Synopsis text file, and reduce the number of bullet points to a maximum of 5.

We have done the requested modifications.

We have done the requested task.

18) As part of the EMBO Publications transparent editorial process initiative (see our policy here: https://www.embopress.org/transparent-process#Review_Process), Molecular Systems Biology will publish online a Peer Review File (PRF) to accompany accepted manuscripts. This file will be published in conjunction with your paper and will include the anonymous referee reports, your point-by-point response and all pertinent correspondence relating to the manuscript. Let us know whether you agree with the publication of the PRF and as here, if you want to remove or not any figures from it prior to publication. Please note that the Authors checklist will be published at the end of the PRF.

We agree with the publication of PRF as here.

19) After your paper is published, we may promote it on social media. If you have any handles or hashtags for Bluesky you would like included, please let us know.

My bluesky handle: @matemanc.bsky.social

My X handle: @matemanc

20) Please provide a point-by-point letter INCLUDING my comments as well as the reviewer's reports and your detailed responses (as Word file).

Reviewer #1:

Overall, the responses to my questions are solid and in several places go beyond what was requested. The only place where I see clearly incomplete answer is Major Comment 5, and to a lesser extent the "conceptual" part of Major Comment 1.

Thank you for this helpful feedback and for your positive assessment of our responses.

Point 1: Comparison of AAS vs SBS in a predictive model. To address the redundancy concern, I suggest adding the following sentence: "When we repeated the multivariate models using COSMIC SBS exposures instead of AASs, model performance was not improved and coefficients were less stable, supporting the view that AASs are a robust low-dimensional representation of mutational processes at the amino-acid level".

We have done the multivariate models and incorporated the text in the manuscript as follows:

„In line with expectation, AAS4-dominant samples were associated with a significantly worse survival than AAS2-dominant ones (Figure 6B). The effect remained significant in a multiple Cox regression model after controlling for age and TMB (Figure 6C). Repeating the analysis with COSMIC SBS exposures in place of AASs (see Methods) did not improve model performance (BIC: 1870.2 for the AAS model vs. 1879.9 for the SBS model; two-sided ANOVA $p = 0.11$), supporting the interpretation that AASs provide a robust, low-dimensional representation of mutational processes at the amino-acid level.”

I also suggest adding a sentence in the Introduction or Discussion: "While COSMIC SBS signatures provide a fine-grained description at the nucleotide level, our AASs are intentionally a lower-dimensional, amino-acid-centric abstraction designed to capture properties most relevant to antigen presentation and T-cell recognition, rather than to maximize resolution of mutational processes per se." That would make it clear that you are not trying to replace COSMIC signatures, but to build a function-oriented layer on top of them.

We included the suggested text in the manuscript as follows:

„...Despite significant strides in this area, the link between mutational signatures and cancer immunity remains largely uncharted. While SBS signatures provide a fine-grained description at the nucleotide level, our amino acid substitution signatures (AAS) are intentionally a lower-dimensional, amino-acid-centric abstraction designed to capture properties most relevant to

antigen presentation and T-cell recognition, rather than to maximize resolution of mutational processes per se. By analysing over 9000 cancer exomes, we identified five distinct classes of AASs, each illustrating specific patterns of amino acid mutations on a genomic scale..."

Point 5: Processing-pipeline limitation (proteasome/TAP/trimming) is not explicitly acknowledged in the revised text. Your new analysis strengthens the link between AASs and detected mutated peptides, which is excellent, but PRIME + binding predictions do not fully model the antigen processing pathway. Therefore, your scores are best viewed as relative measures of immunogenic potential, not absolute probabilities of presentation. I suggest adding a short limitation paragraph such as: "Our analyses focus primarily on HLA binding and T-cell recognition potential (PRIME score) and do not explicitly model upstream processing steps such as proteasomal cleavage, TAP transport, or aminopeptidase trimming. As a result, our predictions should be interpreted as relative measures of neoantigen quality rather than exact estimates of presentation probability. This limitation is partly mitigated by our validation against immunopeptidomics datasets, which confirm that AAS context influences the likelihood that specific substitutions are observed among HLA-I-bound peptides (Appendix Figs. S16-S17), but additional processing-aware modeling will be valuable in future work."

Thank you, we included the suggested paragraph in the discussion section.

Reviewer #2:

The authors addressed all my comments and I suggest acceptance of the manuscript in its current form.

Thank you, we are pleased that our revisions have addressed all your comments.

13th Jan 2026

Manuscript number: MSB-2025-13128RR

Title: Five dominant amino acid substitution signatures shape tumour immunity

Dear Dr. Manczinger,

Thank you again for sending us your revised manuscript. We are now satisfied with the modifications made and I am pleased to inform you that your paper has been accepted for publication.

You may qualify for financial assistance for your publication charges - either via a Springer Nature fully open access agreement or an EMBO initiative. Check your eligibility: <https://link.springer.com/journal/44320/how-to-publish-with-us>

Yours sincerely,

Poonam Bheda, PhD
Scientific Editor
Molecular Systems Biology

>>> Please note that it is Molecular Systems Biology policy for the transcript of the editorial process (containing referee reports and your response letter) to be published as an online supplement to each paper. If you do NOT want this, you will need to inform the Editorial Office via email immediately. More information is available here: <https://link.springer.com/partners/embo-press/editorial-policies#Peer%20review>